# Kinematic properties of regions that can involve persistent contrails over the North Atlantic and Europe during April and May 2024

Sina Maria Hofer[1] and Klaus Martin Gierens[1]

[1]Deutsches Zentrum für Luft- und Raumfahrt, Institut für Physik der Atmosphäre, Oberpfaffenhofen, Germany

**Correspondence:** Sina Maria Hofer (sina.hofer@dlr.de)

**Abstract.**

Contrails can last for many hours in the sky if they form in ice supersaturated regions (ISSRs). Contrail formation is possible once the ambient air is sufficiently moist and cold (below $-40\,°C$). Contrails are persistent when the ambient relative humidity with respect to ice is at least $100\,\%$. Cirrus clouds and contrails consist of ice crystals, which move with the wind. Ice supersaturation is an immaterial feature, which does not generally move with the wind. However, the movement of ISSRs relative to the winds is currently unknown. We analyse the movement of ISSRs and the wind using data for aviation weather forecasts, the WAWFOR package, from the German Weather Service (DWD). Our results demonstrate the kinematic differences and similarities of the movement of ISSRs in comparison to the wind. We show that the ISSRs on average move slower than the wind at the same location, the direction of movement is usually quite similar and the distributions of both velocities follow Weibull distributions. The almost identical direction of the movements is beneficial for contrail lifetimes, but the lifetime of contrails is not generally determined by the lifetime of ISSRs. It happens that contrails are blown out of ISSRs with the wind. We assume that our study can be used as a basis for further analyses of the movement of ISSRs. Furthermore, our method of analysis is applicable to other extended features or areas, for instance for areas where aircraft non-$CO_2$ emissions would have a particularly large climate effect.

## 1 Introduction

Climate change is ongoing and the necessity to take actions against it is getting more and more pressing. All sectors have to reduce their share of the overall climate warming, including aviation. Although the latter sector contributes currently only about 3.5% to the radiative forcing of climate, it cannot be exempted from urgent measures because aviation demand increases at a more rapid pace than the fuel need for a unit distance (the specific fuel consumption) gets reduced by commissioning of more efficient engines (Grewe et al., 2021; Lee et al., 2021). A peculiarity of aviation is its relatively strong climate impact induced by non-$CO_2$ emissions and effects. The individual non-$CO_2$ effects from a single flight depend strongly on the actual ambient conditions (synoptic weather situation, ambient cloudiness, position of the sun, etc.) which may open a possibility for climate- and environmentally friendly air transport by explicitly searching for routes where the climate impact is minimal (Matthes et al., 2021; Yin et al., 2023). An important part of climate-aware aviation is the avoidance of those persistent contrails that contribute to the warming of climate. Persistent contrails can form if the so-called Schmidt-Appleman criterion, introduced

by Schumann (1996), is fulfilled and when the flights transect a so-called ice supersaturated region (ISSR, see Gierens et al., 2012, for a review), where the relative humidity with respect to ice $RH_i \geq 100\%$.

The Schmidt-Appleman criterion is often fulfilled once the ambient temperature is a few degrees below $-40°$C. Supersaturation with respect to ice or in short ice supersaturation (ISS) occurs in particularly cold and humid regions in the troposphere and it is primarily found in areas below the tropopause (Spichtinger et al., 2003; Petzold et al., 2020). Both conditions, low temperature and ice supersaturation, are found in regions that are termed cold ice supersaturated regions (CISSRs) by Irvine and Shine (2015). Ice supersaturated regions (ISSRs) arise mainly from large-scale vertical movements where rising air masses containing water vapour cool adiabatically and the relative humidity increases (Gierens and Brinkop, 2012). CISSRs and IS-SRs are mostly the same areas since the Schmidt-Appleman criterion is most often fulfilled in ISSRs. ISS is often connected to slightly stable to neutral stratification because of the increases in the temperature gradient of rising air layers (Gierens et al., 2022).

ISSRs become visible when either natural cirrus clouds form, or when aircraft fly through them because contrails that form in ISSRs are persistent and can remain in the sky for up to several hours. In spite of this close relation, contrails and their ambient ISSRs do not generally have the same lifetime (Bakan et al., 1994) and can also differ in their movement. Individual contrails and cirrus clouds may be driven out of the parental ISSR, which sooner or later terminates their existence when the ice crystals get into subsaturated air and sublimate. That is, contrails and cirrus on one side and ISSRs on the other have different lifetimes. The lifetime of ISSRs themselves is limited by the dynamics of the atmosphere. As soon as downdraft and thus adiabatic warming occurs, the ice supersaturation decreases and disappears eventually. The different lifetimes motivated us to study the differences between the real wind and the motion (pseudo-wind) of ISSRs, or more precisely, CISSRs. A peculiar difference is that ISSRs have an initiation and an end in time, which also marks the end of their motion. In contrast, the wind is ceaseless (c.f. the apt title of an old textbook, "The ceaseless wind" by Dutton (1986)).

For this initial study and the sake of simplicity, we consider horizontal motions only. We use data from a weather forecast model that covers Europe and the North Atlantic and compare the motion of the air to the movement of CISSRs, that is, we compare the wind with a kind of pseudo-wind. Unfortunately, only two months of data were available for this study, April and May 2024. But we have compared the wind distribution for these two months with those from all other months in twelve different years to ascertain, that the two selected months were not exceptional. The paper is organised as follows: First we describe our data source in section 2. Methods of the data analysis are compiled in section 3. Result are presented in section 4 and these are discussed in section 5. At the end we conclude in section 6.

## 2 Data

The German Weather Service (DWD) provides aviation weather forecasts four times per day (WAWFOR data), based on the weather forecast model ICON (Zängl et al., 2015), with hourly temporal resolution. The usual WAWFOR data inform aviation users on temperature, humidity, winds, etc. (WAWFOR Package 1), but for the German D-KULT project (Demonstrator Klima- und Umweltfreundlicher Lufttransport; Demonstration of climate- and environmentally friendly air transport), addi-

tional datasets are produced that inform on the potential to form persistent contrails, i.e. on CISSRs, and on the climate effect
of other gaseous emissions from aviation. The additional information that is used here is in particular

$PPC$: potential of persistent contrails — a binary value $(0/1)$;

$PPC_{\text{prob}}$: probability of persistent contrails; fraction of ensemble forecast members that have $PPC = 1$ — a real number
between 0 and 1.

$PPC$ is computed from the Schmidt-Appleman criterion (SAC) (Schumann, 1996) applying an overall propulsion efficiency
of $\eta = 0.365$ and using temperature and relative humidity from the regular forecast. In order to compensate a low humidity
bias in the forecast (Gierens et al., 2022), situations with values of relative humidity with respect to ice ($RH_i$) in excess of
93% are considered ice (super) -saturated (Hofer et al., 2024). $PPC_{\text{prob}}$ is simply the number of members of the 40-member
forecast ensemble that predict $PPC = 1$, divided by the total number of ensemble members (40); It can thus obtain values
$n/40; n \in \mathbb{N}_0, 0 \leq n \leq 40$. The meteorological data, $PPC$ and $PPC_{\text{prob}}$, are available globally and with higher spatial reso-
lution for the European region (EU nest, $0.0625° \times 0.0625°$, approx. $6.5\,\text{km} \times 6.5\,\text{km}$). The latter are used here in an area from
$23.5°\text{W}$ to $62.5°\text{E}$ and $29.5°\text{N}$ to $70.5°\text{N}$. Altitudes that are most relevant for air traffic, between $8$ and $12\,\text{km}$ and with $0.5\,\text{km}$
increments, are used. They correspond approximately to pressures $p \in (197, 207, 227, 238, 262, 287, 301, 329, 360)\,\text{hPa}$.

$PPC = 1$ thus marks CISSR grid points where persistent contrails are possible, and $PPC = 0$ marks all the rest where
either no contrails at all or merely short contrails can be formed. Only a few percent of ISSRs are not cold enough for contrail
formation, but once a contrail is formed in a CISSR it can later spread into a slightly warmer ISSR (Wolf et al., 2023). Thus,
we often write simply ISSR instead of CISSR. In addition, we use the wind data to compare it with the pseudo-velocity of
ISSRs. The wind is given with its components in zonal ($x$) and meridional ($y$) directions.

The data covers April and May 2024. Due to missing data, 3 days have been excluded: April 30, May 14 and 28. We use
data for each day at the time of analysis 12 UTC $+t$ hours, where $t \in (10, 11, ..., 20)\,\text{h}$.

We admit that an analysis of only two months does not allow to draw general conclusions, but unfortunately, at the time of
the study, more WAWFOR data of the additional sets were not available. It cannot be excluded that the two selected months
have peculiarities in comparison to other seasons and other years. But a comparison of the wind directions and speeds over all
12 months and over 12 years, with months shuffled such that consecutive months are several years apart, did not point to any
peculiarities (see Appendix).

As we will see later, we always examine ISSRs that appear in three subsequent output times of a forecast run. The dataset
contains 19259 of these groups of ISSRs.

## 3 Methods

### 3.1 Identification of ISSR pairs

In order to determine the kinematics of ISSRs we consider $PPC$ and $PPC_{\text{prob}}$ in subsequent hourly outputs of the WAWFOR
data. A pseudo-velocity of these regions is determined numerically in a way analogue to the numerical determination of a wind
speed at a certain time. To determine the pseudo-velocity, $V_{\text{ISSR}}(t_n)$, at a certain point in time, $t_n$ (e.g. at time step $n$), the

positions of the ISSR on the same pressure level at two additional times are needed, say at $t_{n-1}$ and $t_{n+1}$. It is thus necessary to identify ISSRs at these three points in time in respective three outputs of the forecast. This task is solved first.

To identify ISSRs in several subsequent forecast steps (beginning with $t_n$ and $t_{n-1}$), first their size ($N$, number of simply connected grid points, in longitude, latitude, and diagonally with $PPC = 1$) is determined. ISSRs consisting of less than 500 grid points are not considered further. We introduced the limit of 500 ($N \geq 500$) to avoid examining small ISSR patches and to neglect this kind of noise. With 500 grid boxes, the length scale of ISSRs is about 145 km, quite close to that determined from in-situ data (Gierens and Spichtinger, 2000; Spichtinger and Leschner, 2016). For all other ISSRs we calculate an analogue of a centre of mass, here termed centre of probability (COP) in the following way:

$$x_{\text{COP}} = \frac{\sum_{n=1}^{N} PPC_{\text{prob}_n} \cdot x_n}{\sum_{n=1}^{N} PPC_{\text{prob}_n}} \quad \text{and} \quad y_{\text{COP}} = \frac{\sum_{n=1}^{N} PPC_{\text{prob}_n} \cdot y_n}{\sum_{n=1}^{N} PPC_{\text{prob}_n}}. \tag{1}$$

For the calculation, the spherical coordinates (longitude and latitude) are first converted into cartesian coordinates (with the centre of the Earth as the origin), then the COP is determined in the cartesian system, and finally the coordinates are transformed back into spherical coordinates to provide longitude and latitude of each COP.

Thus, each ISSR has one centre of probability in each forecast and pressure level. Note that $PPC = 1$ for all points that belong to a distinct ISSR, while outside it is zero ($PPC_{\text{prob}}$ can be non-zero, even if $PPC = 0$; the sums do not extend over such points). An example will follow later (see section 3.6 ), where the locations of the COPs are shown as red dots in Figure 2. With the centres of probability at hand, Euclidean distances, $d_{i,j}$, of any pairs of COPs can be determined. ISSRs $i, j$ are not identified if their distance exceeds 280 km.

However, for each ISSR at forecast time $t_n$ we take the three closest ISSRs at forecast time $t_{n-1}$ into closer inspection for the identification with the respective ISSR at $t_n$, as long as these are less than 280 km apart. If there are no such candidates at $t_{n-1}$, the ISSR is not used for further analysis.

We assume 280 km as the maximum distance an ISSR can move within one hour since the maximum wind speed in the data is 281 km h$^{-1}$.

A pair of ISSRs in a respective pair of forecasts must not change size $N$ by more than 45%. From the three candidates (the three closest ISSRs), if they fulfil this condition, the partner with the smallest $d_{i,j}$ is selected. That is, a certain ISSR in one forecast hour is recognised in the adjacent forecast hour (previous or next). If the the size condition is not met for the closest one, the second closest ISSR is chosen as a partner of the other time step. If this condition is not fulfilled for the two closest ISSRs, the third closest ISSR is chosen. If the condition is not fulfilled for this one either, no partner can be found in the other time step and this ISSR at time point $t_n$ is not part of our analysis.

It may happen that more than one ISSR at time $t_n$ has the same partner at time $t_{n-1}$. In this case we find a unique recognition via a measure of similarity provided by the Hu-moments, in that we identify the pair having the smallest similarity difference (see Appendix A for more explanation). The other pair or pairs are not considered for the investigation. The same tests are performed between $t_{n+1}$ and $t_n$ and only if an ISSR is recognised in all three forecasts, its kinematics can be determined. The last criterion refers to the fact that we do not consider ISSRs at the edges of our region, as they can distort the statistics. For

this reason, the distance from each COP (at time $t_{n-1}$, $t_n$ and $t_{n+1}$) to the upper, lower, right and left edge of our area (region: $23.5°$W to $62.5°$E and $29.5°$N to $70.5°$N) is calculated. Only the ISSR triples are used where the COPs are at least $500\,\text{km}$ away from the edge at all three times. For these ISSRs we calculate their kinematics, which is described in the next chapter.

## 3.2 Kinematics of ISSRs

To describe the dynamic motion of an ISSR, one needs to determine how fast it moves. This can be done in many ways, but the simplest way is chosen here, that is, the pseudo-velocity of an ISSR at $t_n$ is taken as the Euclidean distance of its COP at $t_{n+1}$ and $t_{n-1}$ divided by the temporal difference:

$$V_{\text{ISSR}_i}(t_n) = \frac{d(i, n+1, n-1)}{t_{n+1} - t_{n-1}}, \qquad \text{where} \tag{2}$$

$$d(i, n+1, n-1) = \sqrt{[x_{\text{COP}_i}(t_{n+1}) - x_{\text{COP}_i}(t_{n-1})]^2 + [y_{\text{COP}_i}(t_{n+1}) - y_{\text{COP}_i}(t_{n-1})]^2}. \tag{3}$$

$V_{\text{ISSR}_i}(t_n)$ is the pseudo-velocity of ISSR $i$ at time $t_n$, $d(i, n+1, n-1)$ is the Euclidean distance of the COPs of ISSR $i$ at the two time steps before and after $t_n$. In our case, the coordinates are given as longitude and latitude. The conversion of distances on a great circle to distances in km is given in Appendix B.

In order to determine the direction of movement of an ISSR at $t_n$, the vector from its COP at $t_{n-1}$ to the COP at $t_{n+1}$ is used. Using the included angle between this vector and the zonal direction, the trigonometric functions and $V_{\text{ISSR}_i}(t_n)$, the Cartesian components $u_{\text{ISSR}}$ and $v_{\text{ISSR}}$ of the ISSR can be determined.

## 3.3 Characterisation of the winds

As mentioned before, the velocity of the movement of the ISSRs and the wind speed are different. Since the wind and its Cartesian components $u_{\text{wind}}$ and $v_{\text{wind}}$ are only predicted for the grid points, the grid point that is closest to the centre of probability of an ISSR is selected. The magnitude of the wind speed is calculated using the components as follows:

$$V_{\text{wind}} = \sqrt{u_{\text{wind}}^2 + v_{\text{wind}}^2}. \tag{4}$$

## 3.4 Angle between the motion of the ISSRs and the winds

The angle, $\delta$, between the direction of motion of the ISSR and the direction of the local wind is the angle between the two vectors $\boldsymbol{V}_{\text{ISSR}}$ and $\boldsymbol{V}_{\text{wind}}$ with the following convention: If it needs an anticlockwise rotation to turn from $\boldsymbol{V}_{\text{ISSR}}$ to $\boldsymbol{V}_{\text{wind}}$, the angle $\delta > 0$, otherwise $\delta \leq 0$. $\delta$ ranges from $-180°$ to $180°$ (or $-\pi \leq \delta \leq \pi$ in radians). In order to retain the sign in this convention, we use the triple product, $\langle \cdot, \cdot, \cdot \rangle$, with the three unit vectors $\hat{\boldsymbol{V}}_{\text{ISSR}}$, $\hat{\boldsymbol{V}}_{\text{wind}}$, and $\hat{\boldsymbol{z}}$. The latter points upward into the vertical direction. A preliminary result is

$$\delta' = \arcsin\left( \langle \hat{\boldsymbol{V}}_{\text{ISSR}}, \hat{\boldsymbol{V}}_{\text{wind}}, \hat{\boldsymbol{z}} \rangle \right). \tag{5}$$

The result is not yet unique, because the $\sin$ function is not unique in the range $[-\pi, \pi]$. Thus, a second piece of information is obtained using

$$\cos \delta'' = \hat{\boldsymbol{V}}_{\mathrm{ISSR}} \cdot \hat{\boldsymbol{V}}_{\mathrm{wind}}. \tag{6}$$

With this, the final result is

$$
\begin{aligned}
\text{for} \quad \cos \delta'' > 0 \quad &: \quad \delta = \delta' \\
\text{for} \quad \cos \delta'' < 0 \quad \text{and} \quad \delta' < 0 \quad &: \quad \delta = -\pi - \delta' \\
\text{for} \quad \cos \delta'' < 0 \quad \text{and} \quad \delta' > 0 \quad &: \quad \delta = \pi - \delta'.
\end{aligned} \tag{7}
$$

### 3.5 Rotation of ISSRs

As ISSRs are extended features, their characterisation with the COP is just the first of a series of conceivable moments. The second moments (inertia in the mechanical analogue) form a covariance matrix from which further characteristics can be derived via its eigenvalues and eigenvectors. The former can be used to compute an eccentricity of the ISSR, while the latter provide the principal axes of an ISSR.

For a given ISSR with $N$ grid points the covariance matrix is

$$
\quad \Theta = \begin{pmatrix} \sum_{n=1}^{N} PPC_{\mathrm{prob}_n} \cdot (x_n - x_{\mathrm{COP}})^2 & \sum_{n=1}^{N} PPC_{\mathrm{prob}_n} \cdot (x_n - x_{\mathrm{COP}}) \cdot (y_n - y_{\mathrm{COP}}) \\ \sum_{n=1}^{N} PPC_{\mathrm{prob}_n} \cdot (x_n - x_{\mathrm{COP}}) \cdot (y_n - y_{\mathrm{COP}}) & \sum_{n=1}^{N} PPC_{\mathrm{prob}_n} \cdot (y_n - y_{\mathrm{COP}})^2 \end{pmatrix}. \tag{8}
$$

The calculation needs distances, not angles. Thus the necessary transformation is done before application of Equation 8. As $\Theta$ is a symmetric matrix, it has real eigenvalues and the eigenvectors are orthogonal. The eigenvector corresponding to the larger eigenvalue, $\boldsymbol{V}_{\mathrm{PA}}$, defines the principal axis (PA) and the other one the minor axis (MA) of the ISSR on the sphere. The direction of the axes may look strange, because they are computed on the sphere; in the Euclidean geometry of a tangential

plane the axes follow the projection of the ISSRs. However, the tangential plane moves as the COPs move, so they are not used to compute rotation rates. The example in Figure 2 in section 3.6 will show the principal axes in red (the extension in orange) and minor axes in yellow. The angle, $\alpha$, between the principal axis at $t_{n-1}$ and $t_{n+1}$ is determined with the same convention as the angle $\delta$ from above: If it needs a anticlockwise rotation to turn $\boldsymbol{V}_{\mathrm{PA}}(t_{n-1})$ onto $\boldsymbol{V}_{\mathrm{PA}}(t_{n+1})$, $\alpha > 0$, otherwise $\alpha \leq 0$. The calculation of $\alpha$ is thus analogue to the calculation of $\delta$:

$\quad \alpha' = \arcsin \left( \langle \hat{\boldsymbol{V}}_{\mathrm{PA}}(t_{n-1}), \hat{\boldsymbol{V}}_{\mathrm{PA}}(t_{n+1}), \hat{\boldsymbol{z}} \rangle \right), \quad \cos \alpha'' = \hat{\boldsymbol{V}}_{\mathrm{PA}}(t_{n-1}) \cdot \hat{\boldsymbol{V}}_{\mathrm{PA}}(t_{n+1}),$ \hfill (9)

and then the distinction of the cases (c.f. Equation 7) yields $\alpha$.

$\alpha$ can be used to define a kind of angular velocity (pseudo-angular velocity) by dividing it by the temporal difference $t_{n+1} - t_{n-1}$:

$$\dot{\alpha} = \frac{\alpha}{t_{n+1} - t_{n-1}}. \tag{10}$$

Angular velocities are features of extended objects and thus not comparable to either the veering of the wind or its rotation, which are local quantities.

## 3.6 An example

An illustration of the methods and how they work is given in figures 1 and 2. The map of Figure 1 shows the domain of the ICON-EU model. The coloured areas are CISSRs ($PPC = 1$) that are sufficiently large according to the conditions set out
above. The different colours mark different individuals that should be identified in the same forecast run one hour earlier and later. An identification in further forecast hours or even in forecasts with different initialisation times may be tried as well for the purpose of forecast verification. This has been done for the D-KULT project, but is not in the scope of the present paper.

The characterisation of an individual CISSR is exemplified in Figure 2. The coloured area again marks an individual CISSR. Its COP is located at the intersection of the coloured straight lines. Note that the COP lies not necessarily within the CISSR if
the latter has a strongly bent shape. The red-orange lines are the major principal and the yellow ones the minor principle axes of the CISSRs.

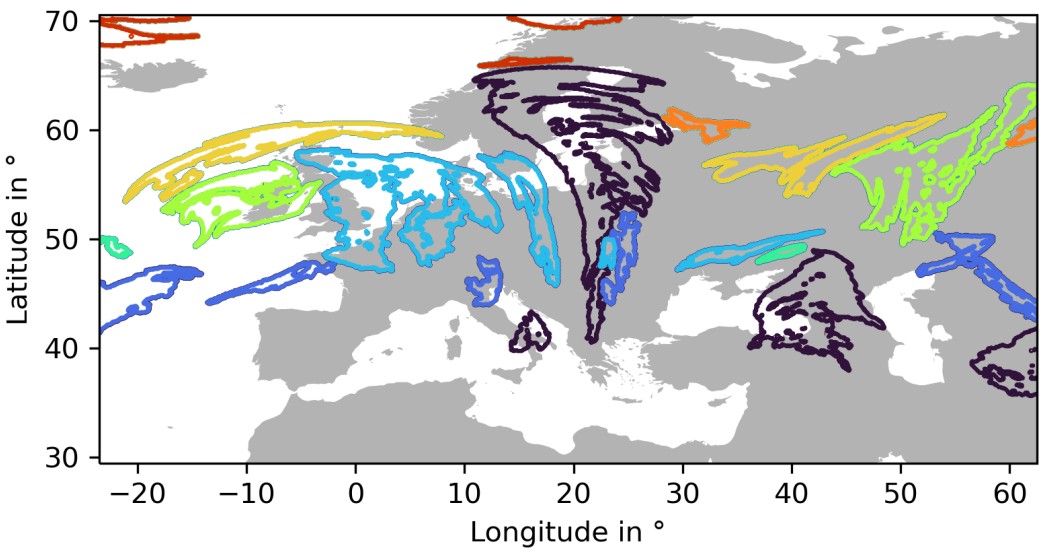

**Figure 1.** Map of the area covered by ICON-EU with regions where $PPC = 1$ (CISSRs) marked in different colours for the example situation from 04.04.2024 at 12 UTC initialisation for $+15$ hours at approximately $301\,\text{hPa}$. These individual CISSRs may be identified in earlier and later forecast hours and in forecasts with differing initialisation time using the methods described in the text. Note: All ISSRs in the area under consideration are shown here. But ISSRs whose COPs are less than $500\,\text{km}$ away from the edge are not included in the statistical analysis.

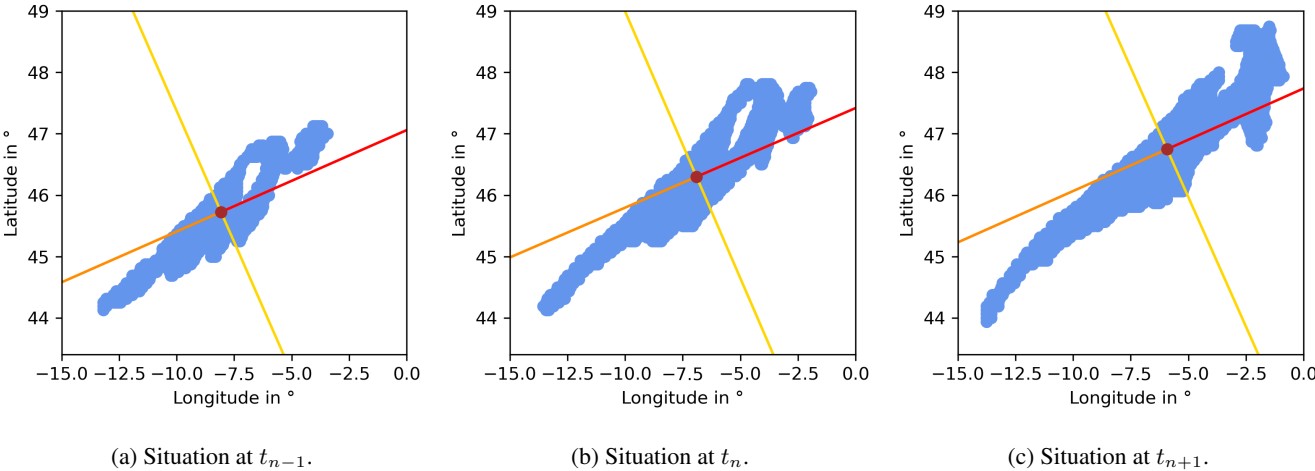

(a) Situation at $t_{n-1}$.  (b) Situation at $t_n$.  (c) Situation at $t_{n+1}$.

**Figure 2.** Basic characteristics of individual CISSRs: The CISSR, that is the grid points with $PPC = 1$ is marked in blue for $t_{n-1}$ (Figure 2a), for $t_n$ (Figure 2b), and for $t_{n+1}$ (Figure 2c). The centre of the crossed lines is the centre of probability (COP). The red-orange lines are the major principal and the yellow ones the minor principle axes of the CISSRs. Note: The CISSR in Figure 2b corresponds to the light blue CISSR in Figure 1 in the southwest between approximately $-15°$ to $0°$ longitude and $43.4°$ to $49°$ latitude. The direction of the axes are computed on the sphere and do therefore not follow the flat projection that is shown in the figure.

## 4 Results

### 4.1 The relative motion of air and embedded ISSRs

#### 4.1.1 Speeds

Figure 3 (left) shows the histogram of pseudo-velocities of ISSRs calculated according to Equation 2. The distribution of ISSR velocities has a mean and a standard deviation of $15.3$ m s$^{-1}$ (black line) and $9.4$ m s$^{-1}$ (blue lines), respectively. The median is $13.5$ m s$^{-1}$ (red line). The distribution is right skewed with a skewness of about $1.1$. The corresponding distribution for the wind speeds at the COPs of the ISSRs, calculated according to Equation 4, are shown in Figure 3 (right). The distribution of wind velocities has a mean and a standard deviation of $20.6$ m s$^{-1}$ (black line) and $10.6$ m s$^{-1}$ (blue lines), respectively. The median is $19.4$ m s$^{-1}$ (red line). The distribution is right skewed with a skewness of about $0.6$. Both distributions can be well approximated by a Weibull distribution (orange), a generalised exponential distribution, which is defined as follows:

$$f(v) = \gamma \cdot m \cdot v^{(m-1)} \cdot \exp\left(-\gamma \cdot v^m\right). \tag{11}$$

The two parameters $m$ and $\gamma$ can be determined by plotting the cumulative distribution functions (CDF) of the pseudo-velocities of ISSRs $V_{\text{ISSR}}$ and the wind speeds $V_{\text{wind}}$ on a so-called Weibull paper (Gierens and Brinkop, 2002), where the x-axis is $\log v$ and the y-axis is $\log(\log(1/(1-CDF)))$. For an illustration see Figure C1 in Appendix C. In this representation, the two CDFs can be approximated by straight lines, which indicates that the original distributions can be approximated very well by Weibull

distributions. The slopes are $m_{\text{ISSR}} = 1.85$ and $m_{\text{wind}} = 1.95$. The intercepts ($s$) are related to $\gamma = e^s$. These intercepts are $s_{\text{ISSR}} = -5.2$ and $s_{\text{wind}} = -6.2$, which are obtained by fitting the two lines to the CDFs in the Weibull paper. Compared to an exponential distribution (a Weibull distribution with $m = 1$) the two wind-speed distributions with their $m > 1$ have quite

different characteristics. At very low $v$ the probability densities approach zero, that is very low wind and very slow motion of ISSRs does hardly occur in the study region. In contrast to an exponential, which maximises probability at zero, the considered wind speeds have mode values (maximum probability) at higher values, namely at $10.9\,\text{m\,s}^{-1}$ for ISSRs and $16.6\,\text{m\,s}^{-1}$ for the wind. Finally, Weibull distributions with $m > 1$ have lower tails than the exponential; this implies that high wind speeds would occur more often than observed if wind speeds were exponentially distributed.

Wind data have been fitted by Weibull distributions already by Dixon and Swift (1984). Weibull distributions of wind speed are used by the wind energy business for the design of their installations and to estimate the expected gain (see, e.g., Wais, 2017; Jung and Schindler, 2019). It seems that Weibull distributions are appropriate to model wind speed distributions both for the altitude of wind turbines and for the upper troposphere, although there is no theoretical justification for this (but read Weibull's remark on this problem, Weibull, 1951). As a test, we have evaluated wind speed statistics at $250\,\text{hPa}$ in the study region for

12 months (every 4th day) in 12 different years (from ERA5 reanalyses, Hersbach et al., 2018), shuffled, such that adjacent months are from years several years apart. For all months, the wind speed statistics closely follows a Weibull distribution (see Appendix D).

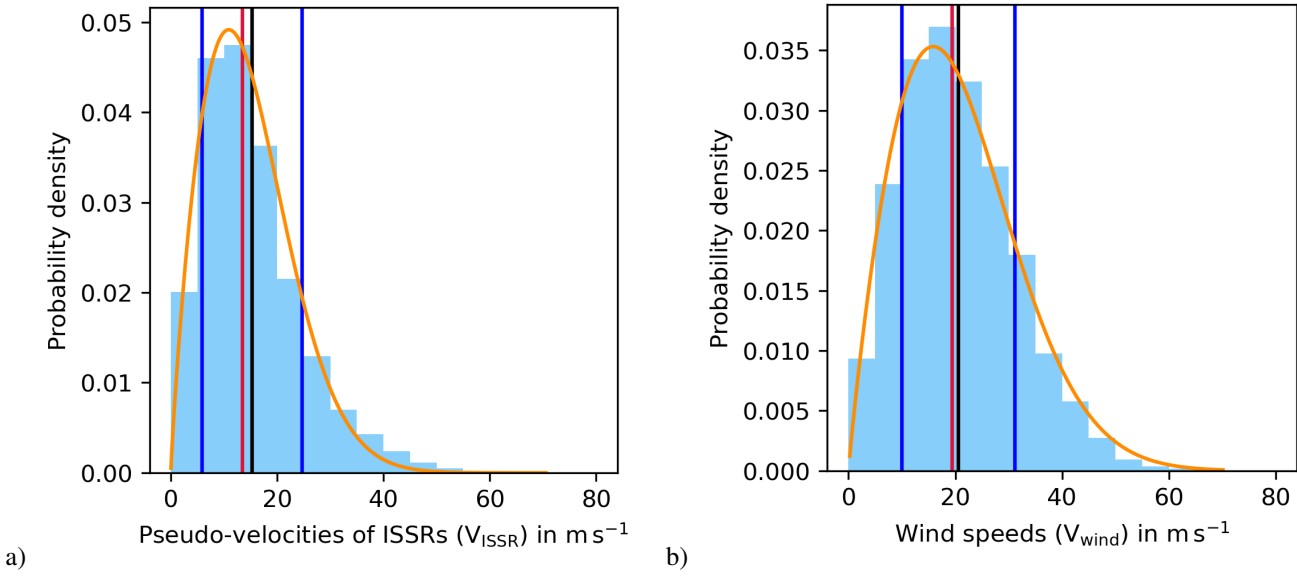

**Figure 3.** Histograms of pseudo-velocities of ISSRs (left) and wind speeds (right) in $\text{m\,s}^{-1}$. The histograms are normalised such that they present approximations of probability density functions, with a resolution of $5\,\text{m\,s}^{-1}$. Means (black lines) $\pm$ standard deviations (dark blue lines) as well as the medians (red lines) are indicated. The orange curves show Weibull distributions that fit the left histogram quite well and the right one excellently. Please note the different scales on the y-axes in both panels.

The distribution of individual differences of pseudo-velocities of ISSRs and of the wind speeds at ISSR COPs is nearly symmetric (the skewness is almost zero, $0.09$) with a peak around $0\,\mathrm{m\,s^{-1}}$, see Figure 4. This means that the velocities of
225 ISSRs and the wind in the COP hardly differ in most cases. The mean of the distribution is $-5.3\,\mathrm{m\,s^{-1}}$ (black) and the standard deviation is $11.8\,\mathrm{m\,s^{-1}}$ (blue). The median of this distribution is $-5.0\,\mathrm{m\,s^{-1}}$ (red). To characterise the peaked shape of the distribution, we use the excess kurtosis, which is the standardised central fourth moment of a distribution minus three (the normal distribution has a kurtosis of three). Positive excess kurtosis values indicate a more pointed and steeper, while negative values indicate a flatter distribution than the normal.

As expected for a peaked distribution, the kurtosis ($1.2$) exceeds zero significantly, if we estimate its uncertainty with $\delta(\mathrm{kurtosis}) \approx \sqrt{(24/N)} \approx 0.035$, where $N = 19259$ is the number of data in the histogram (Press et al., 1989, p. 457). In rare cases the speed differences exceed $50\,\mathrm{m\,s^{-1}}$ in both directions.

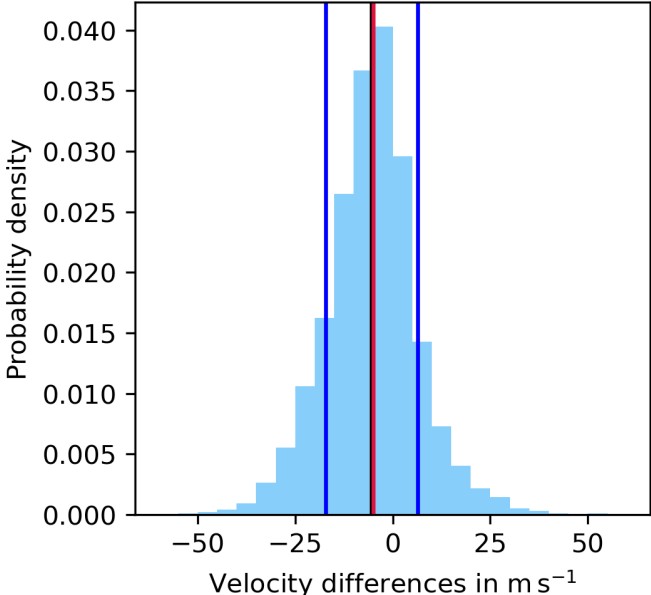

**Figure 4.** Histogram of velocity differences: pseudo-velocities of ISSRs minus the wind speeds in $\mathrm{m\,s^{-1}}$. The histogram is normalised, such that it represents an approximation of the corresponding probability density, with a resolution of $5\,\mathrm{m\,s^{-1}}$. The mean (black line) $\pm$ one standard deviation (dark blue lines) are indicated as well as the median (red line).

Figure 5 shows a joint histogram for ISSR pseudo-speeds and wind speeds. The highest density of cases is at low pseudo-velocities of ISSRs with slightly higher wind speeds. In order to see whether the two (marginal) distributions of the pseudo-
235 velocities of ISSRs and of the wind speeds differ significantly, we perform a Kolmogorov-Smirnov test on the pair of their corresponding cumulative distribution functions ($F_{V_{\mathrm{ISSR}}}$ and $F_{V_{\mathrm{wind}}}$), see Figure 6. To do this, we test the null hypothesis $H_0$ that $F_{V_{\mathrm{ISSR}}} = F_{V_{\mathrm{wind}}}$, which states that the two distributions are equal. The alternative hypothesis $H_1$ is $F_{V_{\mathrm{ISSR}}} \neq F_{V_{\mathrm{wind}}}$ i.e. that the two distribution functions do not match. We select a significance level $\alpha = 5\%$. If $H_0$ is rejected on this level, $H_1$ can

be accepted and the likelihood that this result is wrongly obtained by chance is less than 5%. For the test, the maximum vertical

distance $D$ between the two CDFs ($F_{V_{\mathrm{ISSR}}}$ and $F_{V_{\mathrm{wind}}}$) is determined. The larger this distance, the more likely $H_0$ is rejected. The test statistic $D$ and the $p$-value are calculated using the statistics software package $R$, which yields $D \approx 0.24$ and a $p$-value so much smaller than the significance level, that the exact choice of $\alpha$ does not matter: The null hypothesis $H_0$ can firmly be rejected and the two distributions, $F_{V_{\mathrm{ISSR}}}$ and $F_{V_{\mathrm{wind}}}$, differ significantly from each other.

    Finally, we study the (linear) correlation between the ISSR speeds and the wind speeds. It measures how the individual ISSR

speeds deviate from their mean in a positive or negative direction when simultaneously the wind speeds deviate positively or negatively from their mean and vice versa. It is important to note that the correlation compares anomalies, not the absolute values. This quantity is given as

$$\mathrm{Corr}(V_{\mathrm{ISSR}}, V_{\mathrm{wind}}) = \frac{1}{\sigma_{V_{\mathrm{ISSR}}} \sigma_{V_{\mathrm{wind}}}} \sum_i (V_{\mathrm{ISSR},i} - \langle V_{\mathrm{ISSR}} \rangle)(V_{\mathrm{wind},i} - \langle V_{\mathrm{wind}} \rangle), \tag{12}$$

where the $\sigma_{\mathrm{x}}$ are the two standard deviations and the means are indicated with angular brackets. The sum extends over all our

cases. We find a quite moderate correlation of $0.32$ for the two speed anomalies. This might seem surprising since the absolute speed differences are mainly small. But it simply indicates that the speed differences are positive and negative with similar probability.

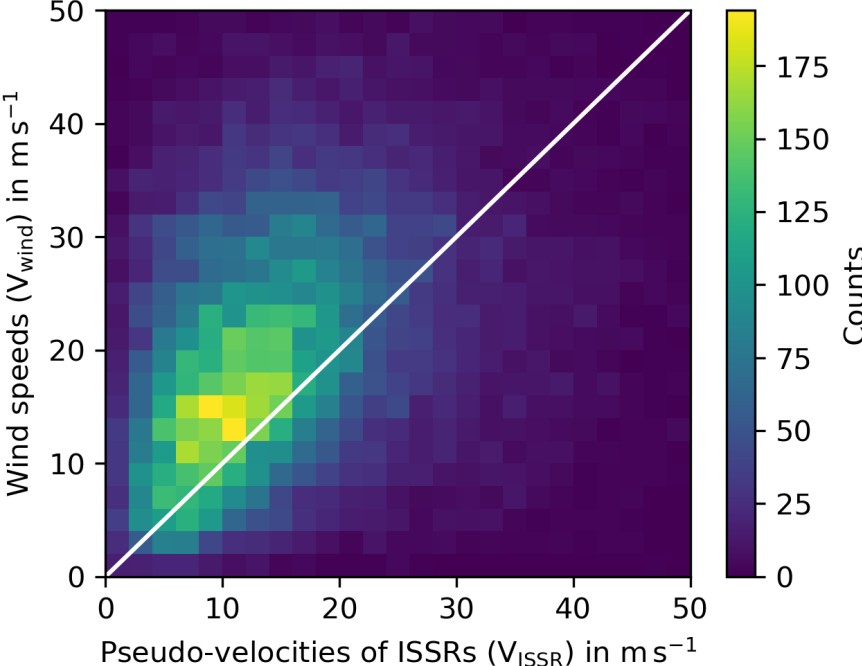

**Figure 5.** Joint histogram of ISSR pseudo-wind speeds and real air wind speeds at the position of the ISSR COPs. The number of joint occurrences in $2\,\mathrm{m\,s^{-1}} \times 2\,\mathrm{m\,s^{-1}}$ square bins are given in the colour bar. The white diagonal marks the points where the two speeds are equal (or nearly so). Higher speeds than $50\,\mathrm{m\,s^{-1}}$ occur in our data, but very rarely.

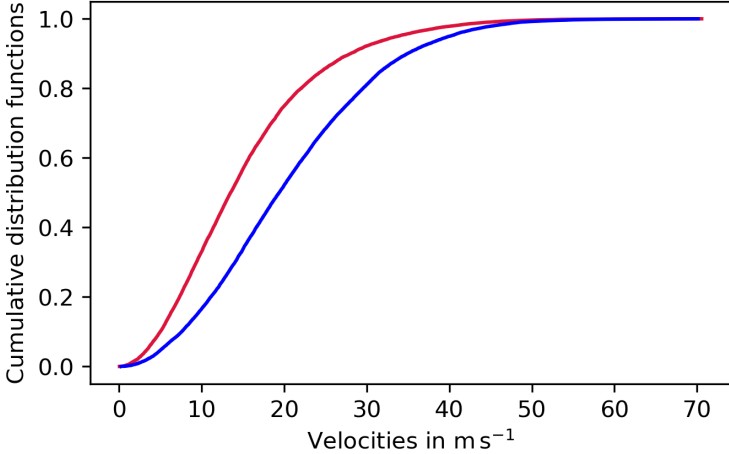

**Figure 6.** Cumulative distribution functions $F_{V_{\mathrm{ISSR}}}$ (red) and $F_{V_{\mathrm{wind}}}$ (blue) .

In some cases, there are significant velocity differences between the motion of ISSRs and the wind (as can be seen in Figure 4 due to the large standard deviation). For the case of the largest speed difference (61.27 m/s), the synoptic situation
was examined more closely (April 4, 2024, at 00 UTC). It turned out that for this specific case the two COPs 1 h before and after $t_n$ are almost on top of each other, while the COP at $t_n$ lies further west of it. Thus, the vector from $t_{n-1}$ to $t_{n+1}$ is also very short, which means that the resulting velocity for the ISSR movement at $t_n$ is very low. The wind, however, blows strongly to the east. In addition, east of the COPs, strong upward movements can be observed, where new air is constantly being supplied from below, which cools and increases its RHi. This means that new ISS is constantly being created (or enlarged) at
this point, while the horizontal wind simply moves on. The gradient of the vertical wind would probably show a maximum at this location. However, this case study only shows the explanation for the case of the maximum speed difference. There are also other cases with significant differences, but not all of them can be examined within the scope of this work. However, this would be very interesting for further analysis.

### 4.1.2   Directions

Next, we consider the direction of the movement of the ISSRs and the wind at their COPs separately, where $0°$ refers to movement to the north and $90°$ corresponds to eastward motion, see Figure 7. (Please note that this convention is different from meteorological use; it is rather oriented on a compass. We use eastward wind instead of west wind, for instance.) Both histograms show that the most frequent direction of motion of ISSRs and of the winds at ISSR COPs is eastward. There is also a second maximum in the westward direction for the direction of ISSR movements, but this is much weaker than the
eastward maximum. The similarity of the histograms does not necessarily imply that the motion of an ISSR and the wind at its COP are often aligned as the histograms show the statistics of both quantities separately. To check this, we consider again

a two-dimensional histogram, that is, the joint probability of the directions of motion (Figure 8). This shows that both ISSRs and the wind usually move in very similar directions, that is, the motions are aligned.

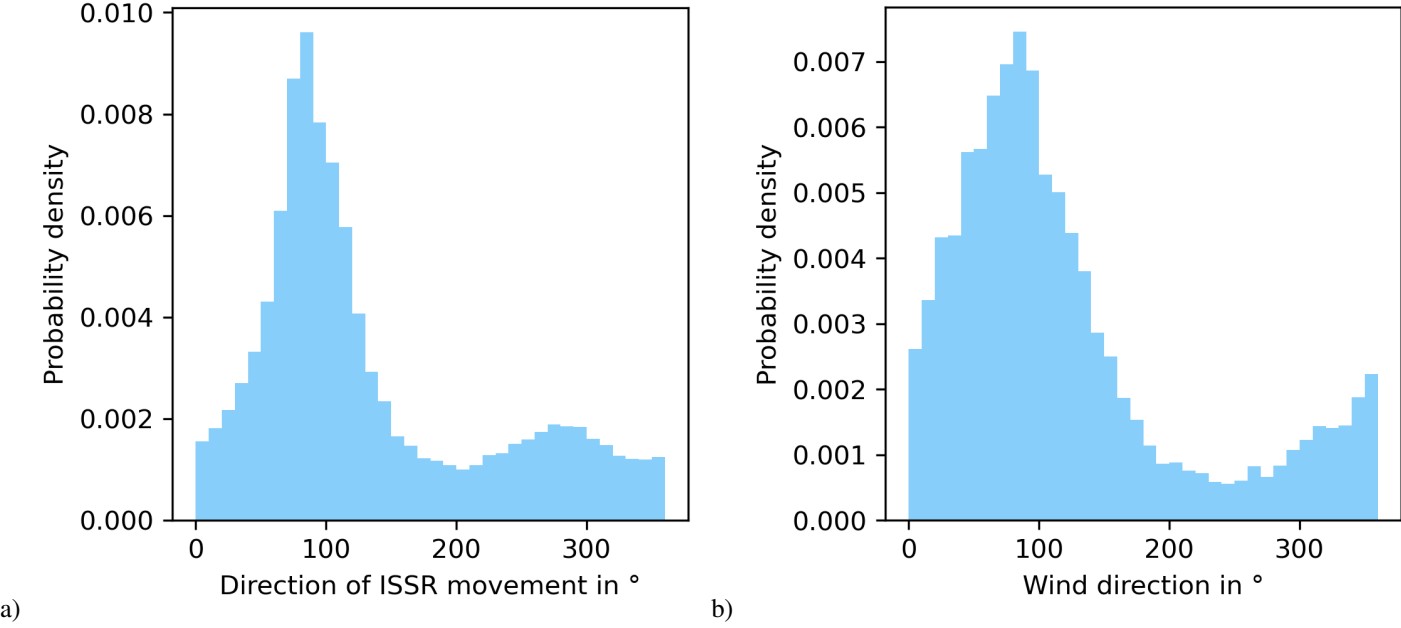

a)  b)

**Figure 7.** Histograms of the direction of movement of ISSRs (left panel) and the wind at their COPs (right panel). Angles range from 0° to 360°, where 0° or 360° means movement to the north, 90° a movement to the east, and so on. The histograms are normalised, such that they represent approximations of the corresponding probability densities, with a resolution of $10°$. The histograms are very similar. Both have their maximum between 80° and 90° degrees, which indicates a predominant west-to-east movement.

Figure 9 shows the differences of both directions of motion. In this histogram, positive angles mean that the wind vector is
in counterclockwise direction from the pseudo-wind vector (e.g. the ISSR moves eastward while the air moves to the north). The mean angle is 3.5° with a standard deviation of 63.7°. The median is 1.9°. If their directions differ then the difference is slightly more common in the anti-clockwise direction than in the clockwise direction, see Figure 8. The peaked shape of the distribution at 0° again shows the quite strong directional alignment of ISSRs and winds. However, the standard deviation is not small and thus cases with quite different directions of motion are no exceptions. The differences in directions and speed
mean that air parcels belonging to an ISSR will sooner or later leave it. Consequences for contrail lifetimes are analysed using trajectory calculations by Hofer and Gierens (2025).

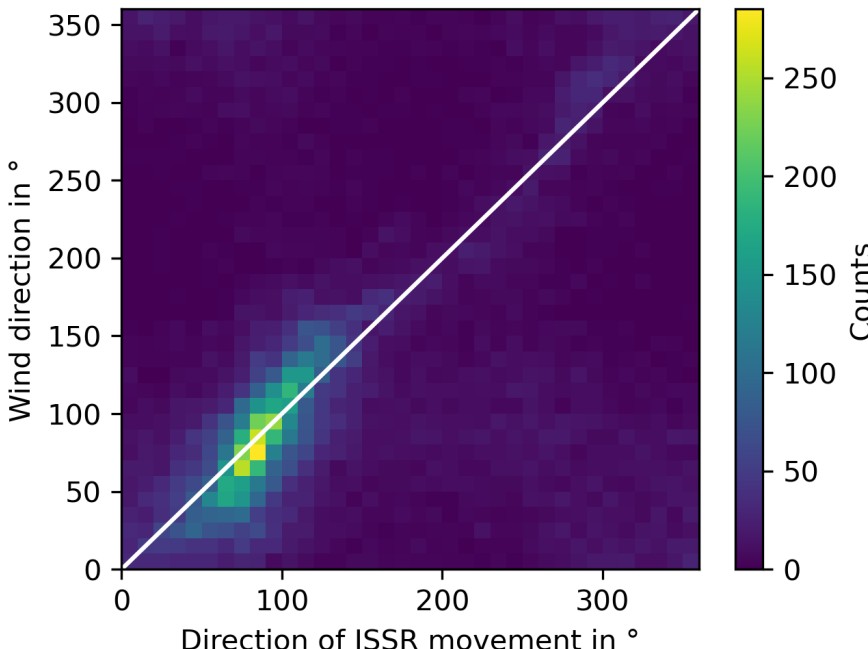

**Figure 8.** Joint probability distribution of the direction of motion of ISSRs and the wind at their COPs. The number of joint occurrences in $10° \times 10°$ square bins are given in the colour bar. The white diagonal marks the points where the two directions are equal (or nearly so). The alignment is quite strong.

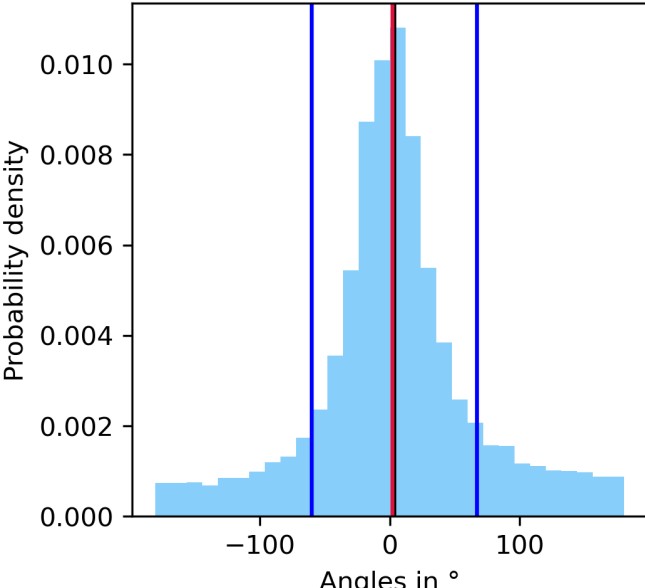

**Figure 9.** Histogram of the angles (in degree) between the movement of ISSRs (vector of COP at $t_{n-1}$ and COP at $t_{n+1}$) and wind vectors from $-180°$ to $180°$. Positive angles mean counterclockwise rotation from the ISSR vector to the wind vector. The histogram is normalised, such that it represents an approximation to a probability density function, with a resolution of $12°$. Mean $\pm$ one standard deviation are indicated with the black and blue lines and the median with a red line, respectively.

## 4.2 Rotation of ISSRs

The rotation of ISSRs is treated as the rotation of its principle axes in time. This can be caused by a real rotation (if the whole system rotates), but a change of the shape or the spatial distribution of probability around the COP can induce rotation of the principle axes as well; thus it is appropriate to speak of a pseudo-rotation with a corresponding pseudo-angular speed.

Figure 10 shows a histogram of the pseudo-angular speed of ISSRs. It is remarkably symmetric around zero (the skewness is almost zero) and peaks at zero. That is, ISSRs most often do hardly rotate and if so, they do it with similar probability in clockwise and anti-clockwise directions. Accordingly, both mean and median are very nearly $0°\,\mathrm{h}^{-1}$ (black and red line) with a standard deviation of $20.5°\,\mathrm{h}^{-1}$ (blue lines).

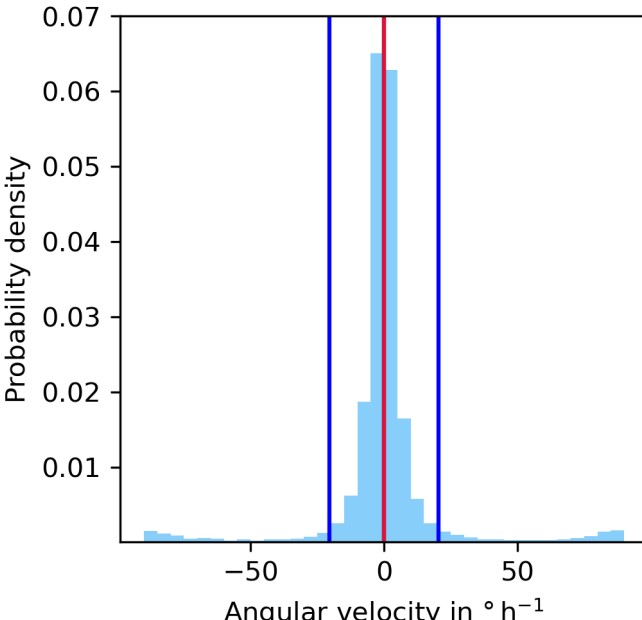

**Figure 10.** Histogram of the differences in angles between the principal axis of $t_{n-1}$ and principal axis of $t_{n+1}$ covered within two hours, i.e. the angular velocity $\dot{\alpha}$ (in $^{\circ}\,\mathrm{h}^{-1}$). The histogram is normalised, such that it represents an approximation of the corresponding probability density, with a resolution of $5^{\circ}\,\mathrm{h}^{-1}$. The mean (black line) $\pm$ one standard deviation (dark blue lines) are indicated as well as the median (red line). Note, that the median and the mean are nearly the same. For this reason the former one can hardly be seen.

## 5  Discussion

### 5.1  Discussion of the methods

This analysis was constrained to horizontal movement only, for the sake of simplicity. This means that the vertical movement of ISSRs was not taken into account. This might seem surprising since vertical lift of moist air is the primary source of ice supersaturation (Gierens et al., 2012). However, vertical movements are two to three orders of magnitudes smaller than horizontal movements in the atmosphere. Therefore we think that this simplification may be justified. Anyway, vertical movements of ISSRs could pretend horizontal movements (e.g. by shape changes) or could perturb the identification of ISSRs in consecutive forecasts if, for example, uplifting air leads to new supersaturation popping up on the considered pressure level close to an ISSR that already exists on that level.

Two ISSRs can also coagulate to a single one or vice versa. In principle, the mechanistic parts of this study (i.e., determination of the COP and the principle axes) can be performed in three dimensions straightforwardly, but the problems with the identification of ISSRs in consecutive forecasts would become much more involved. For instants, application of the Hu moments would not be possible. Therefore, this study constrains the analysis to two dimensions.

ISSRs are not rigid bodies, they generally change shape, extension and total probability while they move and rotate. Thus, one could think of dividing their kinematic properties into internal and external effects, internal being the change in shape and total probability, and external the "true" shift and rotation. Both, internal and external effects change the COP and the principle axes, but it is very difficult and perhaps even impossible or useless to disentangle internal and external effects. One could argue that external effects are caused by shifts and rotation of the carrier fluid air that contains the ISSR, that is, it would be characterised by $V_{\mathrm{wind}}$ and a large-scale rotation of the air. Then the difference $V_{\mathrm{ISSR}} - V_{\mathrm{wind}}$ would be completely due to internal changes of shape, extension and probability distribution of an ISSR.

Considering rotation, the situation is analogous, but also numerically quite difficult. A large scale rotation of the air could be given by the circulation of the wind along the perimeter of an ISSR which, however, is not well defined in a discrete model space. Similarly, the partition of rotation of the ISSR into external and internal cannot be computed adequately since rotation is not well defined in discrete spaces (e.g. Thibault, 2010). Because of these unsolved problems, we have not tried to separate $V_{\mathrm{ISSR}}$ or $\dot{\alpha}$ into external and internal contributions.

We admit that the study is based on only two months of data and it is confined to the EU-nest of the ICON model. As the atmospheric dynamics which is at the base of our findings changes seasonally and geographically, the relation between ISSR motion and the motion of the ambient wind may change from season to season and geographically, in particular in the meridional direction. However, the data necessary to check this become available only in the coming months in the course of the project. As a basic test for seasonal and/or interannual variations we have checked that wind speed statistics follow Weibull distributions with nearly equal exponent over all the selected months (Jan to Dec) and years (2013-2024), at least for the considered study region. From this it seems that April and May 2024 have a quite usual wind speed distribution.

## 5.2 Implications for contrail lifetimes

It turned out that ISSRs move very often with similar speeds and in similar directions as the wind at their COPs do. A similar direction of the movements implies that contrails are not easily blown out of their ambient ISSRs which favours the growth of their ice crystals. In such cases contrails end once their crystals are sufficiently large for sedimentation or once the ice supersaturation itself vanishes by subsidence, increasing temperature and thus decreasing relative humidity. However, contrail lifetimes are not controlled by the lifetime of their parent ISSR for the rare cases where wind and ISSR directions differ. In such cases contrails are blown out of the ISSR with the wind. The reverse process is not possible; persistent contrails cannot be blown into an ISSR because if a contrail does not form within an ISSR, it evaporates quickly so there is nothing left to be blown into the ISSR. Thus, individual contrails must have a shorter lifetime on average than their parental ISSR exists.

This is shown by Hofer and Gierens (2025) who performed trajectory calculations for air parcels which initially reside within an ISSR. The individual air parcels are followed until they eventually either leave the ISSR or until the relative humidity itself falls under ice saturation. This leads to a statistics of "survival" times (again a Weibull distribution) which allows to derive a synoptic time-scale for contrail termination. This time-scale is a few hours, which is consistent with results from satellite tracking of contrails (Gierens and Vázquez-Navarro, 2018).

However, an ISSR may be always filled with new contrails if new aircraft cross it producing new contrails. This has been observed 30 years ago by Bakan et al. (1994).

## 5.3 Application to other features

The presented method has originally been developed for verification of forecasts of CISSRs. An older forecast is to be compared to a newer forecast for the same valid time. The latter forecast serves as a proxy for the reality that is not otherwise available. With the mechanical method we simply determine the COPs and principle axes of individual CISSRs in both forecasts, additionally to their size and total probability, and calculate the individual differences (or other statistics if desired). We note that the method can be used for other features as well, for instance for areas where aircraft non-$CO_2$ emissions would have a particularly large climate effect. For this purpose, we would use the (algorithmic) Climate Change function fields (Dietmüller et al., 2022; Yin et al., 2023), select a threshold (which isolates maxima) and use the value of the aCCF (or the excess over the threshold) in place of the $PPC_{\mathrm{prob}}$. Then everything else is completely analogous to the analyses presented in the current paper.

## 6 Conclusion

In this study, two months of WAWFOR data of DWD are analysed to obtain information about the movement of ice supersaturated regions (ISSRs) and to compare this with the local wind at the ISSRs. To this end we use humidity, temperature and wind information and determine the location of ISSRs. We develop a procedure to identify ISSRs in three consecutive forecasts. For each ISSR we determine a centre of probability (COP) and the principle axes at this point. Speeds of ISSRs are determined by the movement of the COPs together with rotations of their principle axes. For these positions the wind speeds and the rotation of the wind vectors are recorded. These data form the basis of our comparisons.

On average, the wind speeds are larger than the ISSR movement speed. The mean of the pseudo-velocities of ISSRs is approximately $15\,\mathrm{m\,s}^{-1}$. The mean of the wind speeds is around $21\,\mathrm{m\,s}^{-1}$. Both distributions, the ISSR speed and the wind speed, follow a Weibull distribution very well. In most cases ISSRs move at a similar speed as the local wind. An analogous statement pertains to the directions of the movement. Both ISSRs and the local wind move in most cases in eastward direction. Extended ISSRs not only move straight forward but also rotate, but quite slowly. Within one hour the rotation angle rarely exceeds $\pm 10°$. Clockwise and counterclockwise rotations occur with almost equal probability.

The study results imply that persistent contrails, which move with the wind, should often move parallel to and inside their parent ISSR. Under this condition, the lifetime of contrails is constrained by sedimentation of ice crystals or by the end of the lifetime of the ISSR itself. Once ISSRs and the wind move in different directions, contrails are blown out of the ISSR into the subsaturated environment and vanish.

Finally, we note that the methods used in this paper can be applied to other extended features in the atmosphere in an analogous way.

One can determine centre-of-mass and principle axes for any 2-dimensional coherent and closed feature in a weather map with the current methods, at least after suitable adaptations (for instance replacement of $PPC$ and $PPC_{\mathrm{prob}}$). If the same feature can be delineated on, say satellite data, or if the same features are compared in various forecasts (with different time horizons of from different forecast models), there is possibility of comparison and to check the plausibility of the features under consideration quantitatively using the centres-of-mass and the principle axes.

As an example, one could use the algorithmic climate change functions from the WAWFOR-Klima data, set a threshold value and define everything where the threshold is exceeded as a feature. Our method can then be applied. Then one might use the 40 ensemble members of ICON to see how these features behave over the whole ensemble to check whether the considered forecast produces a coherent or incoherent picture of the situation.

*Code availability.* Python codes can be shared on request.

*Data availability.*

## Appendix A: Hu-moments

Hu-moments are based on normalised central moments. The general formula of normalised central moments in our application is:

$$\eta_{k,l} = \frac{\sum_x \sum_y PPC_{\mathrm{prob}}(x,y) \cdot (x - x_{\mathrm{COP}})^k \cdot (y - y_{\mathrm{COP}})^l}{[\sum_x \sum_y PPC_{\mathrm{prob}}(x,y)]^{1 + \frac{k+l}{2}}}. \tag{A1}$$

All $\eta_{k,l}$ whose $k + l = 2$ are related to the entries of the covariance matrix $\Theta$ (see Equation 8).

Normalised central moments are invariant to translation and scale. The advantage of the seven Hu-moments is that they are invariant to translation, scale and rotation. The first six Hu-moments are also invariant to reflection; however, for the seventh Hu-moment the sign changes for reflection. The Hu-moments can be calculated this way (Prokop and Reeves, 1992; Huang and Leng, 2010; Mallick and Bapat, 2018):

$$h_1 = \eta_{20} + \eta_{02} \tag{A2}$$

$$h_2 = (\eta_{20} - \eta_{02})^2 + 4 \cdot \eta_{11}^2 \tag{A3}$$

$$h_3 = (\eta_{30} - 3 \cdot \eta_{12})^2 + (3 \cdot \eta_{21} - \eta_{03}^2) \tag{A4}$$

$$h_4 = (\eta_{30} + \eta_{12})^2 + (\eta_{21} + \eta_{03})^2 \tag{A5}$$

$$h_5 = (\eta_{30} - 3 \cdot \eta_{12}) \cdot (\eta_{30} + \eta_{12}) \cdot [(\eta_{30} + \eta_{12})^2 - 3 \cdot (\eta_{21} + \eta_{03})^2] + \tag{A6}$$

$$+ (3 \cdot \eta_{21} - \eta_{03}) \cdot (\eta_{21} + \eta_{03}) \cdot [3 \cdot (\eta_{30} + \eta_{12})^2 - (\eta_{21} + \eta_{03})^2] \tag{A7}$$

$$h_6 = (\eta_{20} - \eta_{02}) \cdot [(\eta_{30} + \eta_{12})^2 - (\eta_{21} + \eta_{03})^2 + 4 \cdot \eta_{11} \cdot (\eta_{30} + \eta_{12}) \cdot (\eta_{21} + \eta_{03})]) \tag{A8}$$

$$h_7 = (3 \cdot \eta_{21} - \eta_{03}) \cdot (\eta_{30} + \eta_{12}) \cdot [(\eta_{30} + \eta_{12})^2 - 3 \cdot (\eta_{21} + \eta_{03})^2] + \tag{A9}$$

$$+ (\eta_{30} - 3 \cdot \eta_{12}) \cdot (\eta_{21} + \eta_{03}) \cdot [3 \cdot (\eta_{30} + \eta_{12})^2 - (\eta_{21} + \eta_{03})^2]. \tag{A10}$$

Since the seven Hu-moments have different orders of magnitude and are therefore difficult to compare, they are adjusted using a log transformation:

$$H_i = -\text{sgn}(h_i) \cdot \log(|h_i|). \tag{A11}$$

These log transformed $H_i$ are then combined and a difference can be calculated that reflects the similarity difference between two forms of ISSR $A$ and ISSR $B$:

$$D(A, B) = \sum_{i=1}^{7} \left| \frac{1}{H_i^B} - \frac{1}{H_i^A} \right|. \tag{A12}$$

In order to identify two shapes of ISSR $A$ and ISSR $B$ in subsequent forecasts, $D(A, B)$ should be as small as possible.

## Appendix B: Distances on great circles

The distance between two points with longitudes $\lambda_{1,2}$ and latitudes $\varphi_{1,2}$ on a sphere can be determined using the equations for an orthodrome, i.e. the shortest connection between two points on a sphere (see any textbook on navigation or spherical geometry and trigonometry). Let us assume that $\lambda_{1,2}$ and $\varphi_{1,2}$ are given in radians. Let $\Delta_\lambda = |\lambda_1 - \lambda_2|$. Then the distance, $d_{1,2}$, on the surface of the Earth is given as.

$$d_{1,2} = R \arccos\left[\sin(\varphi_1)\sin(\varphi_2) + \cos(\varphi_1)\cos(\varphi_2) \cdot \cos(\Delta_\lambda)\right], \tag{B1}$$

with the radius of the Earth $R \approx 6373$ km. In case, that the longitudes and latitudes are given in degrees, they must be translated into radians with the factor $\pi/180°$.

**Appendix C: Cumulative distribution functions of the pseudo-velocities of ISSRs and the wind speeds plotted on Weibull paper**

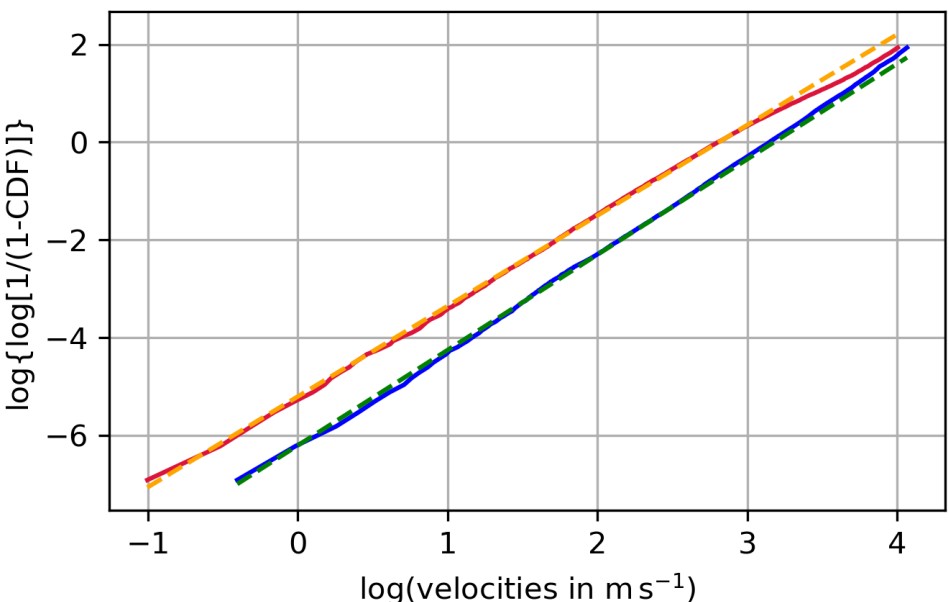

**Figure C1.** Cumulative distribution functions of the pseudo-velocities of ISSRs $V_{\mathrm{ISSR}}$ (red) and the wind speeds $V_{\mathrm{wind}}$ (blue) plotted on Weibull paper together with linear fits as dashed lines: $g_{\mathrm{ISSR}} = 1.85 \cdot x - 5.2$ (orange) and $g_{\mathrm{wind}} = 1.95 \cdot x - 6.2$ (green).

**Appendix D: Comparison of the wind speed and direction distributions with those of other months and years**

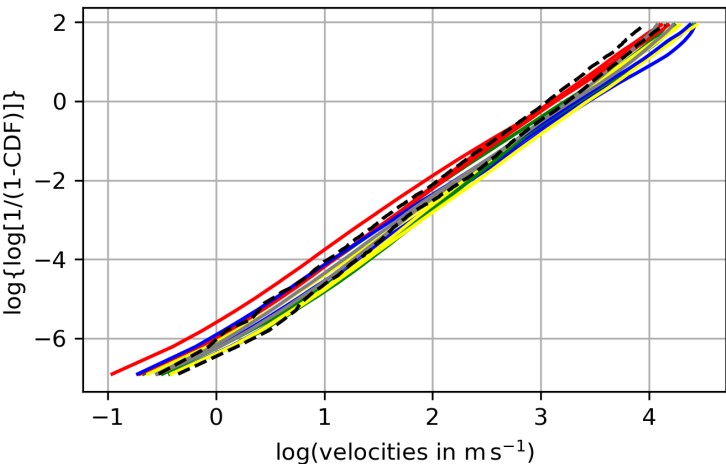

**Figure D1.** Cumulative distribution functions of the winds within the ISSRs of the two analysed months April and May 2024 as dashed black lines plotted on a Weibull probability diagram together with cumulative distribution functions of all winds of April and May 2024 as grey lines and the winds of 12 months of 12 years, shuffled (spring in the Northern Hemisphere in green, summer in red, fall in yellow and winter in blue).

To test that the two analysed months, April and May 2024, do not have exceptional wind conditions, we additionally analysed the horizontal winds on 250 hPa in the study region for 12 months from 12 years (January 2017, February 2014, March 2019, April 2022, May 2015, June 2024, July 2021, August 2018, September 2023, October 2016, November 2013, December 2020). The distribution of wind speeds are shown in Figure D1. It shows the cumulative distribution functions (cdf) of the two analysed months April and May as dashed black lines plotted on a Weibull probability diagram together with cumulative distribution functions of the selected 12 months (every 4th day, with spring in the Northern Hemisphere in green, summer in red, fall in yellow and winter in blue). The CDFs of April and May 2024 are in the middle of the others and show no particular outliers. All lines show a very similar slope, only their intercepts differ slightly. Those of the warmer months/seasons tend to have higher intercepts than those of the colder months/seasons. The Weibull pdf (probability density function) is relatively flat if the intercept is low (higher variance, as in winter), and it gets sharper and more peaked with increasing intercept (lower variance, as in summer). Additionally, we have checked the wind-roses for the two investigated months and the other 12 months of our test data. By and large they are similar and there is no indication that April and May 2024 had any kind of exceptional wind distribution (speed and direction).

*Author contributions.* This paper is part of SH's PhD thesis. SH wrote the codes, ran the calculations, analysed the results and produced the figures. KG supervises her research. Both authors discussed the methods and results and wrote the paper.

*Competing interests.* The authors declare no competing interests.

*Acknowledgements.* This research contributes to and is supported by the project D-KULT, Demonstrator Klimafreundliche Luftfahrt (Förderkennzeichen 20M2111A), within the Luftfahrtforschungsprogramm LuFo VI of the German Bundesministerium für Wirtschaft und Klimaschutz. This work used resources of the Deutsches Klimarechenzentrum (DKRZ) granted by its Scientific Steering Committee (WLA) under project ID bd1357. The authors would like to thank Annemarie Lottermoser for her thorough reading and commenting a draft manuscript.

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
