# Peer review of "Kinematic properties of regions that can involve persistent contrails over the North Atlantic and Europe during April and May 2024"

_EGUsphere, 2024_

## Referee Comment (RC1)

**Comments on "Kinematic properties of regions that can involve persistent contrails" by Sina Maria Hofer and Klaus Martin Gierens (https://doi.org/10.5194/egusphere-2024-3520)**

This paper uses two months of data from the German Weather Service (DWD) aviation weather forecasts (WAWFOR) created with ICON during the D-KULT project. Specifically, the manuscript aims to derive the kinematics of the center of probability (COP) of ice supersaturated regions (ISSRs) and the wind speed at these points.

The paper first describes the data set used and then explains how ISSRs are identified and their respective COPs are determined. To monitor the movement and speedy of ISSRs, the change in position at the location of the ISSR is determined by identifying pairs of ISSR between three consecutive times. In addition, the extension (in the horizontal direction) is determined by the number of consecutive model grid points that are flagged for ISSR. The horizontal extent is determined and a covariance matrix is used to calculate the rotation of the ISSR. This information is then used to determine the ISSR kinematics in terms of speed, direction, and rotation, but also in relation to the underlying wind field.

The main idea of the paper, is to infer the advection of ISSR by the wind field, which is much more accurately to predict than the ISSR themselves. This might provide insight into the life time and spreading of contrails. While this idea is of general interest for mitigating contrail formation and spreading, where information about the location of ISSR is needed, there are several major comments that have to be addressed before this paper can be considered for publication in ACP. The specific major and minor comments are listed below.

**Major comments:**
(1) The entire study is based on two months of data from the D-KULT project. While I understand that dedicated simulations are limited in time, I see major problems with this short time span. Investigating only two months does not even cover a single season of the year, and therefore does not account for changes in global and regional circulation patterns that can occur throughout the year. Even more problematic is the general possibility that these two months and the derived ISSR kinematics may be influenced by an atypical wind field. This would render the investigation and numbers unusable. The authors should at least provide an overview of the general wind pattern in the area (direction, pressure systems) during the two-month period and compare this period with, for example, the 10 year monthly averages for April and May.
While the problem of representation is briefly mentioned in lines 262 to 266, it should be made clear in the introduction and summary.

(2) Follow-up to major comment (1): Even if the two months are representative for an average April and May, these two months, as mentioned in (1), cannot be considered as representative for a longer period of time, nor for the entire Earth. In this respect, the title of the manuscript is too general and promises more than the study can deliver. Therefore, I suggest two options:
(i) The authors explicitly mention the investigated time frame and the specific region in the title, e.g., "Kinematic properties of regions that may involve persistent contrails over the

North Atlantic and Europe during April and May."
or
(ii) present the two months of data as a test case for their proposed method of inferring ISSR kinematics. This would also require a rewording of the title, e.g., "A proposed method to infer kinematics of ISSR applied to two months of aviation weather forecast data."
At the very least, it must be clear that the conclusions given in the study are limited to a very short time period and a specific region, and are not as general as the title suggests.

(3) Throughout the manuscript, several statistical parameters are determined, while the information gained from the statistical tests and parameters is not contextualized with existing literature or used later in the text. For example, in lines 177 and following, the Weibull distribution is introduced and used to fit the wind speed distribution. However, nowhere in the text is the Weibull distribution compared to existing literature, nor is it stated what the information gained from the fit can be used for. The authors mention that Dixon and Swift fitted Weibull distributions but what did they to with the information? The authors should continue their discussion of the Weibull distribution and what its purpose is, e.g., where this information can be applied. The wind speed distributions are likely sensitive to season and location.
It is also a general deficiency of the manuscript and not limited to the use of the Weibull distributions that mere numbers and statistics are given, but the discussion and conclusion of the analysis is short and potentially interesting ideas are not followed to the end. Perhaps I missed the implications of several statistics, but then the authors should better clarify their individual intentions behind the given statistics and parameters.

(4) Follow-up to major comment (3): Some of the results are not very surprising, maybe even trivial, as the authors themselves admit, for example in line 189. To make the study more informative and to use the potential of the proposed method, it would be good to actually look at the ambient conditions - wind direction, speed, and temperature -  where significant differences between the kinematics of the ISSR (COP) and the wind field appear. Much more can be learned from where and when differences occur than just identifying similarities between COP and the wind field.

(5) The authors explain the identification of ISSR, the COP, and the derived motion of ISSRs. For example, an eastward motion is denoted with an angle of 90°. However, the definition of the wind direction is not clear. In meteorology, wind direction indicates where the wind is coming from. See https://confluence.ecmwf.int/pages/viewpage.action?pageId=133262398 . Considering the investigated area (23.5°W to 62.5°E and 29.5°N to 70.5°N), I assume a primarily westerly wind (wind is coming from the west and moves to the east), meaning a wind direction around 270°. Looking at Figure7(right panel), which has a peak at around 90-100°, one has to assume that the wind direction that is given in the manuscript indicates where the wind is going (wind is going east and defined as 90°). Am I interpreting this correctly? If I am wrong in the assumption, for example because April and May were dominated by easterly winds, then I must apologize. But please check and clarify the definition.

(6) L211-212: If the correlation between the speed of each COP and the wind at the COP is low, does this mean that ISSR simply do not move with the wind, and even move in the opposite direction?  The authors write: "But it simply indicates that the speed differences are positive

and negative with similar probability". Later the authors also write in lines 220-221: " This shows that both ISSRs and the wind usually move in very similar directions, that is, the motions are aligned." This is somewhat contradictory.

(7) L211-212: I find it an interesting feature that ISSR and wind are in opposite directions, but it is not discussed. It would be very informative to read what happens in cases where COP and the wind move in opposite directions. What are the dynamics behind this? In such situations, I would expect the largest discrepancies between prediction and observation / reality when ISSR are assumed to be advected with the wind.

**Minor comments:**

L6: Please clarify what is meant by "**material** ice crystals". Ice crystals are always material objects.

L7: Abbreviate Ice supersaturated regions with "ISSR"?

L8: The use of "their" is not clear in this context. Does it refer to the ice crystals, the ISSR, or the wind field?

L27: "..thermodynamic condition, the so-called Schmidt-Appleman criterion,…" please consider rephrasing this sentence. It raises the impression that Schmidt-Appleman criterion is a thermodynamic condition but it is a combination of several conditions taking into account the temperature and humidity (supersaturation)

L73-74: This is already mentioned in lines 42-43. Please consider removing duplicate information.

L97-98: What do the authors mean by "three candidate partner" since the authors mentioned distances between pairs (two)? Please write more clearly. The authors probably mean the single ISSR appearing in three consecutive time steps?

Subsection 3.3: A separate subsection for calculating wind speed is not needed. It could be combined with Section 3.2. This would also be the part wind direction definition should be added. (see major comment 6).

Section 3.6: The example could already be used during the introduction of the different metrics and parameters. This would make definitions more illustrative and better to understand. The authors might consider this in a revised version of the manuscript.

Figure2: The major and minor axes determined in panels (a) to (c) look rotated by a certain -α. From the text, I would assume that one of the major axes should align closely with the longest extension of the ISSR (blue region).

Figure 3 and 7: The authors might consider adding (a) and (b) to the left and right panels of the plots, respectively.

Figure3 caption: The authors may want to write "standard deviations (dark blue lines)" as it might be confused with the blue bars. The authors may also compare the fit of the Weibull distributions with the measurements by saying "quiet well" and "excellent". Please specify "quiet well" and "excellent".

Figure 3 and Figure 5 could be combined into a single plot by using marginal distribution plots, i.e., plotting Fig3 (left) along the x-axis of Fig5 and Fig3 (right) along the y-axis of Fig5. The same holds for Figures 7 and 8.

Figure4: While I do not think there is much additional information in this plot beyond what is already written in the text (lines 186-192), I suggest limiting the x-range to, for example, an interval of [-60,60]. Then the marginal deviation from 0 ms$^{-1}$ velocity difference might become visible. In addition, the choice of colors should be reconsidered. The red line on light blue is very similar to the dark blue lines. This might be problematic for color blind people people or when printed in b/w.

L172: "…,respectively." is missing at the end of the sentence.

L175: "…,respectively." is missing at the end of the sentence.

L186: What do the authors mean by "real wind speeds"? Is there a difference between wind speed and real wind speed?

L186-190: For a distribution that closely resembles a normal distribution (no skewness) shouldn't the peak of the distribution be close to the mean and median? Is the shift in the peak simply due to the selected bin size of the distribution?

L201: $p < 2.2*10^{-16}$ is this a reasonable number to give? I would assume that this number is already close to numerical accuracy.

L204: "…, but they are real." What is the intention of this phrase? The authors applied to Kolmogorov-Smirnov test to determine whether the distributions are similar. According the authors calculation, the test was negative and the hypothesis of equal distributions was rejected. So the distributions are different, and there is no need to further convince the reader.

L227: Please remove "as stated above" or specify what the authors are referring to.

L227: What is meant by "a real rotation"?

L242-243: The authors only consider for horizontal motion. One reason given is ".. another one can appear in that vicinity, so that this one is interpreted as the actual ISSR". But this incorrect identification could also happen on a vertical level.?

L295: "Pseudo-velocities" in lowercase.

L305: This brings up a new point not really discussed before. Could the authors further elaborate on how they would use their proposed method to validated forecasts? What would the authors compare in case of such a validation?

---

## Author Response (AR1)

**Replies**

Sina Hofer and Klaus Gierens

**Answers to Reviewers**

We thank the reviewers for the comments. For convenience, we repeat the comments and then give our replies, which are printed in italics.

*First, we want to mention, that we noticed in our evaluation, that ISSRs also appeared on the edges of the region under consideration, which can distort the statistics and analyses if, for example, an ISSR extends beyond the area. For this reason, we introduced a $500\,km$ safety margin and did not take the ISSRs near the edges into account in the evaluation. We therefore reevaluated our data, which is why the figures and numbers in the manuscript have changed slightly; the basic statement of the paper has not changed.*

**Review 1:**

This paper uses two months of data from the German Weather Service (DWD) aviation weather forecasts (WAWFOR) created with ICON during the D-KULT project. Specifically, the manuscript aims to derive the kinematics of the center of probability (COP) of ice supersaturated regions (ISSRs) and the wind speed at these points. The paper first describes the data set used and then explains how ISSRs are identified and their respective COPs are determined. To monitor the movement and speedy of ISSRs, the change in position at the location of the ISSR is determined by identifying pairs of ISSR between three consecutive times. In addition, the extension (in the horizontal direction) is determined by the number of consecutive model grid points that are flagged for ISSR. The horizontal extent is determined and a covariance matrix is used to calculate the rotation of the ISSR. This information is then used to determine the ISSR kinematics in terms of speed, direction, and rotation, but also in relation to the underlying wind field. The main idea of the paper, is to infer the advection of ISSR by the wind field, which is much more accurately to predict than the ISSR themselves. This might provide insight into the life time and spreading of contrails. While this idea is of general interest for mitigating contrail formation and spreading, where information about the location of ISSR is needed, there are several major comments that have to be addressed before this paper can be considered for publication in ACP. The specific major and minor comments are listed below.

**Major comments:**

1. The entire study is based on two months of data from the D-KULT project. While I understand that dedicated simulations are limited in time, I see major problems with this short time span. Investigating only two months does not even cover a single season of the year, and therefore does not account for changes in global and regional circulation patterns that can occur throughout the year. Even more problematic is the general possibility that these two months and the derived ISSR kinematics may be influenced by an atypical wind field. This would render the investigation and numbers unusable. The authors should at least provide an overview of the general wind pattern in the area (direction, pressure systems) during the two-month period and compare this period with, for example, the 10 year monthly averages for April and May. While the problem of representation is briefly mentioned in lines 262 to 266, it should be made clear in the introduction and summary.

   *REPLY: Thank you for the comment! We are aware that the data set is not generally valid for the whole year. At this time, we only had this data and from June onwards there was a change in the data. However, we only wanted to use the data before the change so that the months are comparable and that we avoid comparing apples with oranges. The introduction has now a sentence at the end that states that only two months are considered.*

   *We checked the large-scale circulation (german: Grosswetterlagen) for the two months and found, apart from the fact, that the wind came rarely from the east, that a description of the large-scale circulation would need at least one additional page with text that distracts from the actual topic. This is a problem since the class of Grosswetterlage changes*

about every three days, and it is not fruitful to report this in a paper on ISSR kinematics. We tried as well an objective circulation classification offered by DWD, but without success, because this has 40 classes, that is, more classes than days per month, so the results for a single month are too noisy for robust results.

To show that the two analysed months in the manuscript are not months with exceptional meteorological conditions, we additionally analysed the horizontal winds on 250 hPa in the study region for 12 months from 12 years (January 2017, February 2014, March 2019, April 2022, May 2015, June 2024, July 2021, August 2018, September 2023, October 2016, November 2013, December 2020). The distribution of wind speeds are shown in Figure 8. It shows the cumulative distribution functions (cdf) of the two analysed months April and May as dashed black lines plotted on a Weibull probability diagram together with cumulative distribution functions of the selected 12 months (every 4th day, with spring in the Northern Hemisphere in green, summer in red, fall in yellow and winter in blue). The cdfs of the two months originally analysed in the manuscript are in the middle of the others and show no particular outliers. All lines show a very similar slope, only their intercepts differ slightly. Those of the warmer months/seasons tend to have higher intercepts than those of the colder months/seasons. The Weibull pdf (probability density function) is relatively flat if the intercept is low (higher variance, as in winter), and it gets sharper and more peaked with increasing intercept (lower variance, as in summer).

The wind directions of the two months used for the paper and the selected 12 months are displayed as wind roses in Figure 7. The winds of each month are shown individually, divided into the 16 main directions. The numbers on the x-axis show how often a particular direction occurs overall (the direction in which the wind blows). The colour coding symbolizes the average speed of all winds in one of the 16 directions. The two months analysed in the paper (04.2024 and 05.2024) do not differ from the other months and in particular not from the other April and May from other years (04.2022 and 05.2015). There is only one outlier among the other months (01.2017) in terms of speed. Everyone pretty much agrees on the direction (mostly E, otherwise ENE).

[Figure]

[Figure]

[Figure]

[Figure]

[Figure]

[Figure]

[Figure]

[Figure]

[Figure]

[Figure]

[Figure]

**Figure 7.** Windroses of the two months used for the paper and the selected 12 months.

[Figure]

**Figure 8.** Cumulative distribution functions of the two analysed months April and May in the manuscript (u and v at the positions of ISSRs) as dashed black lines plotted on a Weibull probability diagram together with cumulative distribution functions of April and May 2024 of the whole analysed region in grey and the 12 months of 12 years, shuffled (spring in the Northern Hemisphere in green, summer in red, fall in yellow and winter in blue).

2. Follow-up to major comment (1): Even if the two months are representative for an average April and May, these two months, as mentioned in (1), cannot be considered as representative for a longer period of time, nor for the entire Earth. In this respect, the title of the manuscript is too general and promises more than the study can deliver. Therefore, I suggest two options:

(i) The authors explicitly mention the investigated time frame and the specific region in the title, e.g., Kinematic properties of regions that may involve persistent contrails over the North Atlantic and Europe during April and May. or

(ii) present the two months of data as a test case for their proposed method of inferring ISSR kinematics. This would also require a rewording of the title, e.g., A proposed method to infer kinematics of ISSR applied to two months of aviation weather forecast data. At the very least, it must be clear that the conclusions given in the study are limited to a very short time period and a specific region, and are not as general as the title suggests.

*REPLY: We agree and will change the title as suggested in the first option. The second option is not what we intend, since it will be difficult to apply our method to other data, where usually $PPC$ and $PPC_{prob}$ are not both available.*

3. Throughout the manuscript, several statistical parameters are determined, while the information gained from the statistical tests and parameters is not contextualized with existing literature or used later in the text. For example, in lines 177 and following, the Weibull distribution is introduced and used to fit the wind speed distribution. However, nowhere in the text is the Weibull distribution compared to existing literature, nor is it stated what the information gained from the fit can be used for. The authors mention that Dixon and Swift fitted Weibull distributions but what did they to with the information? The authors should continue their discussion of the Weibull distribution and what its purpose is, e.g., where this information can be applied. The wind speed distributions are likely sensitive to season and location. It is also a general deficiency of the manuscript and not limited to the use of the Weibull distributions that mere numbers and statistics are given, but the discussion and conclusion of the analysis is short and potentially interesting ideas are not followed to the end. Perhaps I missed the implications of several statistics, but then the authors should better clarify their individual intentions behind the given statistics and parameters.

*REPLY: There are several questions in this comment. Let us therefore begin with the basic one: why does one use distributions to describe data?*

*If one has to deal with a large amount of data, it is generally neither possible nor interesting to keep all the values in*

*mind or at least to remember the shape of a histogram. This is the fundamental reason why data are described statistically, typically using simple measures like mean values and standard deviations. In this way, information is condensed and can be kept in mind and, in particular, it is more general than single values. We think, your first comment goes into a similar direction where you ask for statistics of more than just two months.*

*The next question is, why we use a Weibull distribution and whether this has any implications. We add in the paper two references from the literature of the wind energy business (Wais, 2017; Jung and Schindler, 2019). According to these papers, wind speed distributions can be fitted to a number of analytical expressions (e.g. Gamma distribution, etc.), but the Weibull distribution is by far the most used one (more than 90% of the papers). Although this literature deals for obvious reasons with the wind in about 100-200 m altitude, it seems that the Weibull distribution does also excellently fit the wind in the upper troposphere.*

*By the way, we have here another important reason why analytical expressions (distributions) are used to model wind speeds: they can be used relatively easily in models for the expected energy gain from a wind turbine.*

*We have no answer to the question why just the Weibull distribution is so useful to describe wind speed distributions. The answers that are reasonable in other circumstances do not fit for the wind. Neither is there a reliability context that could be applied, nor a differential equation whose solution is the Weibull distribution (as in nearest neighbour statistics). An interpretation via extreme-value statistics (where the Weibull d. is one of the three limiting distributions) is neither possible. So this point must be left open. All that we can contribute here is a quote from Weibull's original paper: "The objection has been stated that this distribution function has no theoretical basis. But in so far as the author understands, there are — with very few exceptions — the same objections against all other df [distribution functions], applied to real populations from natural or biological fields, at least in so far as the theoretical basis has anything to do with the population in question. [. . . ] It is believed that in such cases the only way of progressing is to choose a simple function, test it and stick to it as long as none better has been found." (Weibull, 1951).*

*Additional consequences of the Weibull distribution are now added in section 4.1.1.*

4. Follow-up to major comment (3): Some of the results are not very surprising, maybe even trivial, as the authors themselves admit, for example in line 189. To make the study more informative and to use the potential of the proposed method, it would be good to actually look at the ambient conditions - wind direction, speed, and temperature - where significant differences between the kinematics of the ISSR (COP) and the wind field appear. Much more can be learned from where and when differences occur than just identifying similarities between COP and the wind field.

*REPLY: We find a maximum velocity difference between the ISSR movement and the wind of 61.27 m/s. This was recorded on April 4, 2024, at 00 UTC ($t_n$) at an altitude of 300.9 hPa. The COPs at the three time steps are located (longitude, latitude) as follows:*

- *for $t_{n-1}$: −9.24, 47.69,*
- *for $t_n$: −9.77, 47.49 and*
- *for $t_{n+1}$: −9.15, 47.75.*

*The situation is shown in Figure 9. The figure on the left illustrates the entire area and the one on the right the section around the COPs (dark purple: at $t_{n-1}$ and $t_{n+1}$; light violet: at $t_n$). The green contours describe $PPC = 1$ areas. In the left figure, only the wind arrows within the $PPC = 1$ areas are shown, while in the right figure, the wind arrows are shown throughout the entire section. The black contour lines show the normalised geopotential height, and the red and blue areas illustrate the vertical velocity (in blue: upward movement; in red: downward motion). The two COPs 1 h before and after $t_n$ are almost on top of each other, while the COP at $t_n$ lies further west of it. Thus, the vector from $t_{n-1}$ to $t_{n+1}$ is also very short, which means that the resulting velocity for the ISSR movement at $t_n$ is very low. The wind, however, blows strongly to the east. It is also noticeable that west of the COPs, the air moves downward, while east of these, strong upward movements can be observed. This indicates that at the point of steep upward movement, new air is constantly being supplied from below, which cools and increases its RHi. This means that new ISS is constantly being created (or enlarged) at this point, while the horizontal wind component simply moves on. The gradient of the vertical wind would probably show a maximum at this location. However, this case study only shows the explanation for the*

*case of the maximum speed difference. There are also other cases with significant differences, but not all of them can be examined within the scope of this work. However, this would be very interesting for further analysis.*

[Figure]

**Figure 9.** Synoptic situation on April 4, 2024, at 00 UTC at 300.9 hPa where the maximum velocity difference between the ISSR movement and the wind occurred in our data (left figure: entire region; right figure: the section of interest). The figures show the COPs (dark purple: at $t_{n-1}$ and $t_{n+1}$; light violet: at $t_n$), $PPC = 1$ regions as green contours, the normalised geopotential height as black contours, the vertical velocity (in blue: upward movement; in red: downward motion) and wind vectors (left figure: winds only within $PPC = 1$ regions; right figure: winds in the entire section).

5. The authors explain the identification of ISSR, the COP, and the derived motion of ISSRs. For example, an eastward motion is denoted with an angle of $90°$. However, the definition of the wind direction is not clear. In meteorology, wind direction indicates where the wind is coming from. See https://confluence.ecmwf.int/pages/viewpage.action?pageId= 133262398. Considering the investigated area ($23.5°$W to $62.5°$E and $29.5°$N to $70.5°$N), I assume a primarily westerly wind (wind is coming from the west and moves to the east), meaning a wind direction around $270°$. Looking at Figure 7 (right panel), which has a peak at around $90$–$100°$, one has to assume that the wind direction that is given in the manuscript indicates where the wind is going (wind is going east and defined as $90°$). Am I interpreting this correctly? If I am wrong in the assumption, for example because April and May were dominated by easterly winds, then I must apologize. But please check and clarify the definition.

   *REPLY: We are sorry for the confusion and we clarify how we define the wind direction in the revised version (in the subsection "Directions"). In our study, the angle in degrees indicates the direction where the wind is blowing to, not from. We imagine a compass, which is marked from 0 to 360 degrees with North at $0°$ and East at $90°$. The tip of the compass needle points to the direction where the wind goes. Therefore, words like westerly or easterly are not used in the paper, only westward and eastward.*

6. L211-212: If the correlation between the speed of each COP and the wind at the COP is low, does this mean that ISSR simply do not move with the wind, and even move in the opposite direction? The authors write: But it simply indicates that the speed differences are positive and negative with similar probability. Later the authors also write in lines 220-221: This shows that both ISSRs and the wind usually move in very similar directions, that is, the motions are aligned. This is somewhat contradictory.

   *REPLY: Let us explain this a little bit. It is easy and there is no contradiction at all. It is important to know exactly what "correlation" is about. Correlation measures whether ANOMALIES of the wind and the speed of the COP tend to be*

*both positive/negative or not. This is clear from the formula for the correlation coefficient, which does not involve the product of the two winds but instead the product of the wind anomalies, that is, the respecitve differences from the mean values. Thus, the mean values (that is, the dominating flow) are taken out of the calculation and the correlation is about anomalies or fluctuations. In contrast, the histograms (pdfs and cdfs) deal with the values of the wind and the COP speed directly, that is, they do not consider anomalies.*

*It is thus no contradiction at all to have the wind and the COP moving with similar speed and in similar directions while their correlation is weak. Similarly, there would be no contradiction to have a large correlation between quantities a and b, while the mean of a is 1000 and the the mean of b -1000.*

*In the paper, we try to make this clearer by including the word "anomalies" at several occasions.*

7. L211-212: I find it an interesting feature that ISSR and wind are in opposite directions, but it is not discussed. It would be very informative to read what happens in cases where COP and the wind move in opposite directions. What are the dynamics behind this? In such situations, I would expect the largest discrepancies between prediction and observation / reality when ISSR are assumed to be advected with the wind.

*REPLY: Sorry, this sentence is about speed differences and they are sometimes positive and sometimes negative. The word "differences" is already in the paper and there is nothing to add.*

*Negative speed differences do no imply different directions. This is exactly the reason why the correlation is weak.*

**Minor comments:**

L6: Please clarify what is meant by material ice crystals. Ice crystals are always material objects.

*REPLY: We agree and delete the word material. We used it in the manuscript to stress the difference between the motion of matter and the motion of an immaterial feature.*

L7: Abbreviate Ice supersaturated regions with ISSR?

*REPLY: Done*

L8: The use of their is not clear in this context. Does it refer to the ice crystals, the ISSR, or the wind field?

*REPLY: This sentence has been changed.*

L27: ..thermodynamic condition, the so-called Schmidt-Appleman criterion, please consider rephrasing this sentence. It raises the impression that Schmidt-Appleman criterion is a thermodynamic condition but it is a combination of several conditions taking into account the temperature and humidity (supersaturation)

*REPLY:*

*Although we delete the word "thermodynamic, we do not agree. The Schmidt-Appleman criterion is indeed a purely thermodynamic criterion. It is the statement, that while the exhaust gases are mixed with ambient air, that water vapour must achieve liquid (super-) saturation to allow the vapour to condense and droplets to form. We do not completely understand your question, but it seems that you mix up "thermodynamic" with "temperature". The S.A.C. is not a temperature criterion, it is a thermodynamic criterion.*

L73-74: This is already mentioned in lines 42-43. Please consider removing duplicate information.

*REPLY: We believe, you mean lines 32-33. Yes, the sentence in L73-74 can be deleted. Done.*

L97-98: What do the authors mean by three candidate partner since the authors mentioned distances between pairs (two)? Please write more clearly. The authors probably mean the single ISSR appearing in three consecutive time steps?

*REPLY: We have changed the sentence and we hope, it is clearer now, what we do. In principle, we look for the three ISSRs at both $t_{n-1}$ and $t_{n+1}$ that are closest to a certain ISSR at $t_n$. These three are candidates for identification, but the whole identification process is more complex and involves application of a similarity measure (via the Hu moments). So there are three (or less) candidates, and if one of them has the highest similarity to the ISSR under consideration at $t_n$, it is identified with the latter. This is done for all ISSRs at $t_n$.*

Subsection 3.3: A separate subsection for calculating wind speed is not needed. It could be combined with Section 3.2. This would also be the part wind direction definition should be added. (see major comment 6).

*REPLY: Yes, this could be done, but we prefer to leave it as it is. To our opinion readability of the paper does not get improved if distinct informations are stored in the same subsection.*

Section 3.6: The example could already be used during the introduction of the different metrics and parameters. This would make definitions more illustrative and better to understand. The authors might consider this in a revised version of the manuscript.

*REPLY: We are not sure whether such a change would really help. Without the definitions, the example lacks meaning and the reader would ask what these coloured patches are, why they have different colours, what the meaning of the point in the middle is and what this strange cross is. All this becomes clear only after the definitions are given. It might help understanding, if there were references to these figures right at the points where the definitions are given. Such references are provided now.*

Figure2: The major and minor axes determined in panels (a) to (c) look rotated by a certain -a. From the text, I would assume that one of the major axes should align closely with the longest extension of the ISSR (blue region).

*REPLY: This turned out to be a tough question. However, in principle the answer is simple. Usually the tensor of inertia is defined in cartesian geometry, however, the ISSRs lie on the surface of a sphere. This makes the principle axes looking strange if the ISSRs are oblong and oblique. In thinking about the question, we introduced a tangential plane around the COP with the x-axis in the zonal direction and the y-axis perpendicular. The local z-axis goes upward from the centre of the Earth through the COP. In such a plane the calculation of the principle axes occurs in Euclidean geometry and the result looks familiar, see the figure.*

*For the calculation of rotation rates we retain our original method (with the ISSRs on the sphere and not projected on (different) planes). The justification for this is, that the motion of the COP implies the change of the tangential plane from hour to hour, such that it would be necessary to compare directions in different spaces. This is not possible. As the sphere and its coordinate system is fixed, we do the calculation there. Corresponding explanations are given in the revised paper. Minor programming errors have been spotted as well and have been corrected.*

[Figure]

**Figure 10.** The ISSR of Figure 2 in the manuscript projected on the tangential plane with the COP as the origin of the coordinate system (black dots) together with the corresponding major (red) and minor (orange) principle axes. It is seen, that in Eucledian geometry the axes behave as expected.

Figure 3 and 7: The authors might consider adding (a) and (b) to the left and right panels of the plots, respectively.
*REPLY: Done.*

Figure3 caption: The authors may want to write standard deviations (dark blue lines) as it might be confused with the blue bars. The authors may also compare the fit of the Weibull distributions with the measurements by saying quiet well and excellent. Please specify quiet well and excellent.

*REPLY: The standard deviation lines are now characterised as dark blue lines, as suggested. However, we cannot follow the suggestion to specify well and excellent. This is just a qualitative description and to our view this is sufficient. Of course, one*

235 *could easily calculate $\chi^2$ for the two panels. This would just give two numbers, a smaller one (excellent) and a larger one (well). We doubt that such an exercise delivers any valuable knowledge.*

Figure 3 and Figure 5 could be combined into a single plot by using marginal distribution plots, i.e., plotting Fig3 (left) along the x-axis of Fig5 and Fig3 (right) along the y-axis of Fig5. The same holds for Figures 7 and 8.

*REPLY: Indeed this could be done, but it would not improve the paper.*

240 Figure 4: While I do not think there is much additional information in this plot beyond what is already written in the text (lines 186-192), I suggest limiting the x-range to, for example, an interval of [-60,60]. Then the marginal deviation from 0 ms-1 velocity difference might become visible. In addition, the choice of colors should be reconsidered. The red line on light blue is very similar to the dark blue lines. This might be problematic for color blind people people or when printed in b/w.

*REPLY: This is a good idea that we follow.*

245 L172: ,respectively. is missing at the end of the sentence.

*REPLY: Corrected.*

L175: ,respectively. is missing at the end of the sentence.

*REPLY: Corrected.*

250

L186: What do the authors mean by real wind speeds? Is there a difference between wind speed and real wind speed?

*REPLY: We delete the word real.*

L186-190: For a distribution that closely resembles a normal distribution (no skewness) shouldn't the peak of the distribution be close to the mean and median? Is the shift in the peak simply due to the selected bin size of the distribution?

255 *REPLY: Please be aware that zero skewness is not sufficient for being a normal distribution. Many distributions are symmetric. But we can agree on the statement that for a mono-modal symmetric distribution peak, mean and median have the same value. In our figure, the two lines for mean and median are both located in the maximum bin, which is just below $0$.*

[Figure]

**Figure 11.** Histograms of velocity differences for different bin sizes: pseudo-velocities of ISSRs minus the wind speeds in $\mathrm{m\,s^{-1}}$. The histograms are normalised, such that it represents an approximation of the corresponding probability density, with a resolution of $10\,\mathrm{m\,s^{-1}}$ (left) and $2.5\,\mathrm{m\,s^{-1}}$ (right). The mean (black line) $\pm$ one standard deviation (dark blue lines) are indicated as well as the median (red line).

L201: $p < 2.20^{-16}$ is this a reasonable number to give? I would assume that this number is already close to numerical accuracy.

*REPLY: This is true and it might suffice if we just state that the Null Hypothesis can be rejected on any reasonable significance threshold.*

L204: , but they are real. What is the intention of this phrase? The authors applied to Kolmogorov-Smirnov test to determine whether the distributions are similar. According the authors calculation, the test was negative and the hypothesis of equal distributions was rejected. So the distributions are different, and there is no need to further convince the reader.
*REPLY: Ok, we delete this sentence.*

L227: Please remove as stated above or specify what the authors are referring to.
*REPLY: Ok, we delete this phrase.*

L227: What is meant by a real rotation?
*REPLY: If ISSRs were solid bodies, it would be superfluous to specify rotation as real rotation. However, ISSRs aren't solid bodies and rotation of its principle axes can be caused as well by changes of the shape, size, and by changes in the $PPC_{prob}$-field, as explained in the paper. So, to our view, the best way to distinguish these "artifical" rotations from a rotation of the whole ISSR (that is one, where neither shape nor size nor probability distribution are involved) is to qualify the solid-body rotation as real rotation. We don't have a better idea, and we think that "solid-body rotation" would be misleading.*

L242-243: The authors only consider for horizontal motion. One reason given is .. another one can appear in that vicinity, so that this one is interpreted as the actual ISSR. But this incorrect identification could also happen on a vertical level.?
*REPLY: What we mean is that, if there is uprising air (adiabatic cooling, rise of relative humidity), new ice-supersaturation can appear on the considered horizontal level, perhaps close to and connecting to an already existing ISSR. This is not an ideal situation.*
*In order to avoid misunderstanding, we have replaced the sentence by a simpler one. We hope, that helps.*

L295: Pseudo-velocities in lowercase.
*REPLY: Corrected.*

L305: This brings up a new point not really discussed before. Could the authors further elaborate on how they would use their proposed method to validated forecasts? What would the authors compare in case of such a validation?
*REPLY: One can determine centre-of-mass and main axes for any 2-dimensional coherent and closed feature in a weather map with the current methods, at least after suitable adaptations (for instance replacement of $PPC$ and $PPC_{prob}$). If the same feature can be delineated on, say satellite data, or if the same features are compared in various forecasts (with different time horizons of from different NWP model), there is possibility of comparison and to check the plausibility of the features under consideration.*
*As an example, we could use the algorithmic climate change functions from the WAWFOR data, set a treshold value and define everything where the threshold is exceeded as a feature. Our method can then be applied. Then we might use the 40 ensemble members of ICON to see how these features behave over the whole ensemble to check whether the forecast on this date produces a coherent or incoherent picture of the situation.*
*This elaboration has been added to the end of the paper.*

**Review 2:**

**GENERAL REMARKS**

The manuscript introduces a new method to investigate the movements of ice-supersaturated regions (ISSR) in the atmosphere. The authors use two months of data, April and May 2024, from the German Weather Service aviation weather forecast package

(WAWFOR) to analyse the occurrence of ISSRs, determine their Centres of Probability (COP) and investigate the movement of the COPs in comparison to the horizontal winds. The analysis is based on a broad application of statistical methods. The two-months case study shows that the ISSRs move largely with the wind fields of embedding air, but with some slight differences.

The topic of the study is of broad interest for the research on contrail occurrence and evolution in the context of aviation climate impacts and fits very well into the scope of the journal. It is carefully designed and presents a new method for tackling this important question. Given the limited data set, the study is an excellent show case for this new method, but the manuscript draws general conclusions which appear to be too broad for this limited data set.

In that context few general questions arise which need to be answered before the manuscript is accepted for publication.

1. The authors should analyse carefully their results with respect to their general applicability, given the limited time span of two months for this analysis. To allow the readers own judgement, the investigated area needs to be visualized, e.g., by adding borders and coastlines to Fig. 1. Then, the general weather situation in April and May over this area needs to be described, including their representativeness for spring conditions in general. Once these facts are in place, the results presented can be put into context.
   *REPLY: Indeed, the selection of only two months can be a problem and does not allow to draw general conclusions. Unfortunately, other data were not available for the study. At least, we tested in reply to the comments whether the wind statistics (as reflected in the Weibull distribution) changes over the years and the seasons. For this, we selected horizontal wind data on 250 hPa from the same study region from ERA5. These are 12 months (every 4th day) from 12 different years, shuffled, such that months that belong to the same season belong to years that are several years apart. As Fig. 1 of this reply shows, all these months have a similar distribution of the wind speeds, characterised by Weibull distributions. Such a result does not indicate exceptional wind conditions. The wind roses in reply 1 Figure 7 show that the wind directions of April and May 2024 (analysed in the manuscript) agree very well with wind directions of other 12 months over the years (January 2017, February 2014, March 2019, April 2022, May 2015, June 2024, July 2021, August 2018, September 2023, October 2016, November 2013, December 2020). Furthermore, we used the European climate bulletins of DWD to find that both selected months were among the warmest on record which could favour ice supersaturation in the UP. Other climate parameters (e.g. precipitation) are probably less indicative of the humidity conditions in the UT and are not mentioned. Also, the "exceptionally" warm conditions seem to be hardly anymore exceptional. The climate bulletins of the DWD show also that the wind over central Europe came rarely from the east, but, unfortunately, it is not possible to elaborate without a major distraction from the actual topic in this paper on the large-scale circulation. Please refer also to the corresponding reply to Reviewer 1, who had similar comments.*

2. The consequences of the presented results need to be discussed in more depth. As one example, the authors present in Fig. 3, that the average speed of ISSRs is slightly lower than that of the wind field around the ISSRs. This finding would mean in consequence, that after a certain time the movements of the ISSR and of the embedding air are decoupled. Furthermore, the weak correlation of 0.28 between the ISSRs pseudo-velocities and the wind velocities indicates a decoupled movement, see Fig. 5. Is this case and if so, what are the consequences?
   *REPLY: This is true. In fact, to elaborate more on the consequences, in particular the decoupling, we have performed trajectory calculations to show the decoupling. This has been put into wider context of the time-scales of the main mechanisms that let contrails dissolve. The analysis is described in another paper by the same authors (Hofer and Gierens, 2025).*
   *Indeed, what we already indicated in section 5.2, is shown in the companion paper very clearly. We can define a synoptic time-scale (e-folding time) for contrails by statistical means (essentially again via Weibull-statistics) which turns out to be a couple of hours. A corresponding sedimentation time-scale is of the same order, and the combined time-scale for contrail termination is the harmonic mean of the two time-scales. This study shows, for instance, that LES-simulations of contrails overestimate contrail life-times if they don't regard synoptic developments. There are further consequences for mitigation options that lead to reduced soot emission, lower ice crystal numbers and thus to reduced sedimentation*

*time-scales. As the new paper is available as an egusphere preprint, it is not necessary to repeat its results in detail. However, we insert a few linking sentences into section 5.2 with reference to the new paper.*

3. In the same, direction, Fig. 6 clearly shows that the cumulative distribution function of ISSR velocities reaches the 100% plateau value at a significantly lower velocity than the wind speeds. Wind direction and direction of ISSR movement are much closer aligned as is shown in Figs. 7 and 8. A discussion of the consequences of these findings for the movement of ISSR with respect to the embedding air masses would be very helpful.

   *REPLY: While your interpretation of Fig. 6 is correct, that of figs. 7 and 8 is not. Figs. 7 and 8 show that two distributions are similar, but that does not imply that pairs of wind-speeds and ISSR-speeds or the corresponding directions are similar in single cases. This is stated in the paper in sect. 4.1.2: "The similarity of the histograms does not necessarily imply that the motion of an ISSR and the wind at its COP are often aligned as the histograms show the statistics of both quantities separately." The individual velocity differences are shown in Fig. 4 and the individual direction differences in Fig. 9. These figures show that the motions are not always aligned. We follow your suggestion and elaborate more on these issues in section 4.1.2 where we also point to the trajectory calculations (Hofer and Gierens, 2025).*

**SPECIFIC COMMENTS**

1. On line 90, the authors state a limit of 500 grid points as a lower limit for an ISSR to be considered in their analysis. It would be useful to get an idea of the related size in km distance.

   *REPLY: The distance between two grid points is approximately $6.5\,km$. With 500 grid points once can estimate a typical length scale as $\sqrt{(500 \times 6.5 \times 6.5)}\,km \approx 145\,km$, which nicely fits the length scale that Gierens and Spichtinger have derived long ago (Gierens and Spichtinger, 2000; Spichtinger and Leschner, 2016). This simple estimate will be added in the paper.*

2. On line 93, the term $PPC_i$ is used, please define this quantity.

   *REPLY: $PPC_i$ refers to the value for $PPC$ of every grid point, indexed with $i$. But it seems that the index is not actually necessary and we will delete it.*

3. Equation (8) introduces the covariance matrix $\Theta$ which contains the information about the POC of the ISSR and the principal axes. Figure 2 shows respective results for an exemplary ISSR and the resulting principal axes. However, the position of the COP is understandable given the shape of the ISSR but the principal axes are counter-intuitive. A discussion on of the results is requested.

   *REPLY: Please see the long answer given to Reviewer 1 to the same problem (Minor comments, Figure 2).*

4. Line 190: The term kurtosis should be explained for non-specialists in the applied statistical methods.

   *REPLY: Kurtosis is a normalised and centralised fourth moment of a distribution. It describes whether a (monomodal) distribution is flatter or more peaked than a normal distribution. The normal distribution has a kurtosis of three. In order to make the comparison more convenient, the three is usually subtracted from the kurtosis (which is then called excess). We do this as well in the paper.*

   *Kurtosis values above zero indicate a more pointed, steeper distribution, while negative values indicate a flatter distribution. We add some explanatory sentences in the text.*

5. Line 196: It should be stated at the first mention, that FVissr and FVwind describe the cumulative distribution functions.
   *REPLY: Yes, we agree. The information has been supplemented accordingly.*

6. Starting on line 252, the authors discuss the differences in velocities between ISSR and wind and use the analogy of a foehn cloud. However, a foehn cloud is triggered by an orographic feature causing lifting of the overflowing air. Since the orographic feature remains at a fixed position the foehn cloud will stay at the same place while the air is flowing through it. But what is the feature creating an ISSR and why should this feature remain at a fixed point? The explanation given here is not very clear and should be sharpened.

   *REPLY: The idea was to mention a well-known example of a case where wind and feature don't move together. However, it seems to be distracting, therefore we delete this example.*

**MINOR ISSUES**

1. The abstract length needs to be reduced since the journal allows max. 250 words, but the current word count is 299.
   *REPLY: Thank you for the hint, the abstract has been shortened accordingly.*

2. Line 41-42: The use of the terms material for ice crystals and immaterial for ISSRs is difficult to understand. One possible alternative could be to simply write that cirrus and contrails consist of ice crystals moving with the wind, while ISSR is a thermodynamic feature which may move independent of the wind.
   *REPLY: We can understand the confusion here and have deleted the sentence.*

3. Line 62: It should be specified that PPCprob describes a probability.
   *REPLY: Yes, we have added this information to the manuscript.*

4. Line 84: Better use analogue to ?
   *REPLY: Has been changed to "analogue to"*

5. Line 97: Better use defined instead of identified ?
   *REPLY: In this context "identified" is the correct word. Consider that we have ISSRs in several consecutive forecast hours. Our problem is not to define ISSRs, but to determine whether an ISSR that is present at hour $t_n$ is the same as an ISSR that is present at hour $t_{n-1}$. This decision is termed "identification", that is we state that the two ISSRs under consideration are the same (identical). Of course, this has to be understood "cum grano salis" because of the changes with time. We can quote the greek philosopher Heraklites here: $\pi\alpha\nu\tau\alpha\ \varrho\varepsilon\iota$.*

6. Line 118/119: It should be mentioned that uISSRand vISSRare the cartesian components of the horizontal wind vector.
   *REPLY: Done.*

7. Line 252: The term material system for air might be replaced by by shifts and rotations of the carrier fluid air.
   *REPLY: Thank you for this suggestion which we follow.*

8. Line 271: The term However, might be replaced by On the contrary.
   *REPLY: Done.*

9. Line 284/285: The abbreviation aCCF should be introduced.
   *REPLY: aCCF is abbreviation for algorithmic Climate Change functions which was introduced in the same sentence.*

**References**

Gierens, K. and Spichtinger, P.: On the size distribution of ice–supersaturated regions in the upper troposphere and lowermost stratosphere, Ann. Geophys., 18, 499–504, 2000.

Hofer, S. and Gierens, K.: Synoptic and microphysical lifetime constraints for contrails, egusphere-2025-326, 25, 1–23, https://doi.org/10.5194/egusphere-2025-326, 2025.

Jung, C. and Schindler, D.: Wind speed distribution selection – A review of recent development and progress, Renewable and Sustainable Energy Reviews, 114, 109 290, https://doi.org/10.1016/j.rser.2019.109290, 2019.

Spichtinger, P. and Leschner, M.: Horizontal scales of ice-supersaturated regions, Tellus B: Chemical and Physical Meteorology, 68, 29 020, https://doi.org/10.3402/tellusb.v68.29020, 2016.

Wais, P.: A review of Weibull functions in wind sector, Renewable and Sustainable Energy Reviews, 70, 1099–1107, https://doi.org/10.1016/j.rser.2016.12.014, 2017.

Weibull, W.: A statistical distribution function of wide applicability, ASME Journal of Applied Mechanics, pp. 293–297, 1951.

---

## Referee Report (RR1)

**Comments on the second version of "Kinematic properties of regions that can involve persistent contrails" by Sina Maria Hofer and Klaus Martin Gierens (https://doi.org/10.5194/egusphere-2024-3520)**

This is the second time I have reviewed this manuscript after it was revised by the authors. Therefore, a summary of the manuscript is not given here.

I would like to thank the authors for considering and responding to the comments from the first round of reviews. This has improved the understanding of the paper and made the conclusions more accessible to a wider audience, and I agree with the changes made.

---

## Author Response (AR2)

**Replies**

We thank the editor for the comments. For convenience, we repeat the comments and then give our replies, which are printed in italics

- "c.f." should read "cf." it means "compare (with)". Note that often "see" fits better than "cf."

*Reply: We have adapted the manuscript accordingly.*

- "Figure" should be written as "Fig." except at the beginning of a sentence.

*Reply: We have adjusted Fig. and also Sec. and Eq.*

- panels in figure captions should just be referred to as "(a)", "(b)" ... etc. and not as "Figure 2a" (as currently in the caption of your Fig. 2)

*Reply: We changed the references.*

- panels should always be addressed as (a), (b) ... and not as (left) or (right)

*Reply: We have adapted the manuscript accordingly.*

- L254: m/s should read m s^-1 (please check in entire paper)

*Reply: Yes, we changed that.*

- "pseudo-wind": please consider again using this term. I don't like it too much ... there is just a real wind and I am not sure what a "pseudo-wind" should be. What you consider is the propagation speed of the ISSR, so I am fine with "pseudo-velocity of ISSRs" (which you also use in the paper). Although in my view, it could also be just "velocity of ISSRs", not sure why you need "pseudo", but more essential to me would be to avoid "pseudo wind". And if you decide for your favourite term, then I suggest to stick to this term and not switch between pseudo-wind, pseudo-speed, pseudo-velocity ... (e.g., in Fig. 5 it is confusing that the axis label and caption text use different terms).

*Reply: We have deleted pseudo-wind. We only call it pseudo-velocity in the manuscript now.*

- a matter of taste: you have often very short paragraphs (sometimes only one sentence). To me, this makes the fluent reading a bit difficult as some of this single-sentence paragraphs appear not to be very nicely embedded in the flow of the story. For instance the 3 last paragraphs of this paper, don't they belong together? If you agree, you can make minor changes; if not, then fine to leave it as it is since the reviewers did not mention it before.

*REPLY: We have put together some paragraphs.*

- L372: what are "algorithmic climate change functions from the WAWFOR-Klima data"? Please explain or give a reference.

*REPLY: We have added references to the manuscript.*

[Figure]

*The axis designations of Fig. C1 and D1 have also been adjusted.*